# A case for reframing automated medical image classification as segmentation

**Sarah M. Hooper**
Electrical Engineering
Stanford University

**Mayee F. Chen**
Computer Science
Stanford University

**Khaled Saab**
Electrical Engineering
Stanford University

**Kush Bhatia**
Computer Science
Stanford University

**Curtis Langlotz**
Radiology and Biomedical Data Science
Stanford University

**Christopher Ré**
Computer Science
Stanford University

## Abstract

Image classification and segmentation are common applications of deep learning to radiology. While many tasks can be framed using either classification or segmentation, classification has historically been cheaper to label and more widely used. However, recent work has drastically reduced the cost of training segmentation networks. In light of this recent work, we reexamine the choice of training classification vs. segmentation models. First, we use an information theoretic approach to analyze why segmentation vs. classification models may achieve different performances on the same dataset and task. We then implement methods for using segmentation models to classify medical images, which we call *segmentation-for-classification*, and compare these methods against traditional classification on three retrospective datasets (n=2,018–19,237). We use our analysis and experiments to summarize the benefits of using segmentation-for-classification, including: improved sample efficiency, enabling improved performance with fewer labeled images (up to an order of magnitude lower), on low-prevalence classes, and on certain rare subgroups (up to 161.1% improved recall); improved robustness to spurious correlations (up to 44.8% improved robust AUROC); and improved model interpretability, evaluation, and error analysis.

## 1 Introduction

Classification and segmentation are two popular applications of deep learning to radiology [1]. Classification produces coarse-grained, image-level predictions, while segmentation produces fine-grained, pixel-level maps. While many tasks can be framed using either classification or segmentation (Figure 1), classification is often the default framing. For one, classification has been cheaper to label, requiring only one label per image compared to segmentation's tedious, pixel-wise labels. Second, training classification networks to produce image-level outputs mirrors radiologists' standard workflows, where radiologists summarize findings at the image-level and only routinely perform segmentation in a few sub-disciplines that rely on quantitative measurements over the segmentation masks. These factors have led to an abundance of medical image classification models [2, 3]. In contrast, because segmentation has been so cumbersome to label and is performed relatively infrequently by radiologists, segmentation networks are often only trained when the downstream application *requires* it (e.g., for computing the volumes or diameters of structures in an image).

Recent work in label-efficient training enables us to reexamine this paradigm. Self-supervised learning, in-context learning, weakly-supervised learning, and semi-supervised learning can substantially reduce labeling burden [4, 5, 6, 7, 8, 9, 10]. Additionally, more public datasets and broad-use pre-

37th Conference on Neural Information Processing Systems (NeurIPS 2023).

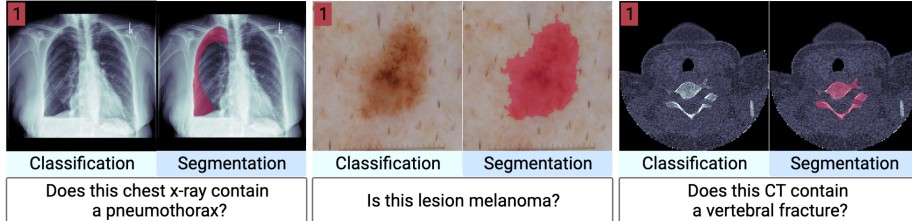

Figure 1: Illustration of three medical image analysis problems that can be framed using either classification or segmentation.

trained networks are coming online [11, 12, 13, 14, 15]. Together, these new methods, datasets, and pretrained backbones are enabling users to develop networks for new segmentation tasks with less and less annotation burden—a trend that will likely continue.

In light of this progress, we reexamine the convention around when to employ segmentation networks. Specifically, we investigate using segmentation networks for medical image classification, exploring if segmentation's fine-grained annotation and output can lead to benefits compared to classification's coarse annotation and output. Our contributions include:

- **Exposition.** We analyze why classification and segmentation networks may perform differently on the same datasets. We show that segmentation leads to more separable and robust embedding spaces, guiding what benefits we should expect to see (Section 2).
- **Best practices.** We implement multiple methods to obtain classification labels from segmentation networks, which we call *segmentation-for-classification* (Section 3). We empirically confirm our analysis (Section 4.1) and compare methods across datasets, tasks, and training conditions to build up best practices for training segmentation-for-classification networks (Section 4.2). We show segmentation-for-classification can improve aggregate performance compared to traditional classification models by up to 16.2% (from 0.74 AUROC to 0.86 AUROC). Finally, we explore semi-supervised segmentation-for-classification, showing existing label-efficient training methods for segmentation can directly benefit segmentation-for-classification.
- **Trade-offs.** We pull together our analysis and experiments to provide consolidated and expanded evidence of the benefits of using segmentation-for-classification, including: improved sample efficiency, enabling improved performance on small datasets, on low-prevalence classes, and on certain rare subgroups (up to 161.1% improved recall); improved robustness to spurious correlations (up to 44.8% improved robust AUROC); and improved model interpretability, evaluation, and error analysis (Section 5).

In summary, conventional wisdom driving the choice between classification and segmentation was formed when labeling burden was a major bottleneck to building neural networks. With recent progress in label-efficient training, we should reconsider conventions. Ultimately, we show that the choice of training a classification vs. segmentation network not only impacts how the model can be used but changes the properties of the model itself, and we show that leveraging segmentation models can lead to higher quality classifiers in common settings. These results indicate segmentation may be a more natural way to train neural networks to interpret and classify images, even though it's not the conventional way humans classify images.

**Related work.** Our study is motivated by past works that employ segmentation networks in classification problems and report various benefits of doing so [16, 17, 18, 19, 20, 21]. A full discussion of related work is included in Appendix A1.

## 2 Analysis

In this section we develop an understanding of why segmentation and classification networks may perform differently given the same dataset and overarching task (i.e., to find a class-of-interest). We cover task specification and notation in Section 2.1 then build up intuition for the benefits of segmentation in Section 2.2.

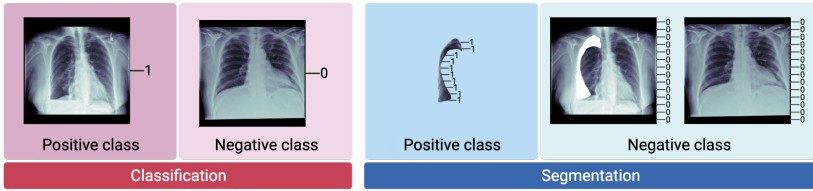

Figure 2: Visualization of the positive and negative class distributions that classification vs. segmentation networks aim to discriminate. In classification, supervision is applied at the image-level; in segmentation, supervision is applied at the pixel-level.

## 2.1 Preliminaries

**Task specification.** We first specify the classification tasks we consider replacing with segmentation. We study classification tasks where the following properties are true: the class-of-interest can be localized in the foreground and does not occupy the entire image; the class-of-interest appears if and only if the classification label is positive; the background should not inform the classification label. Example tasks that meet these specifications are shown in Figure 1. Examples of tasks that don't meet these specifications include classifying the imaging modality or radiographic bone age, where global features of an image inform the class label and there is no clear segmentation target.

**Notation.** We consider a dataset of $N$ images, class labels, and segmentation masks $\mathcal{D} = \{(\boldsymbol{X}^i, y^i, \boldsymbol{S}^i)\}_{i=1}^N$. Each image $\boldsymbol{X}^i$ is made up of $L$ pixels, $\{X_1^i, X_2^i, \ldots, X_L^i\}$. For simplicity of notation, we assume binary classification, so the image-level classification label is $y^i \in \{0, 1\}$ where 1 is the positive label. The segmentation mask $\boldsymbol{S}^i$ contains $L$ pixel-level labels $\{S_1^i, S_2^i, \ldots, S_L^i\}$ indicating which pixels contain the class-of-interest. The image-level class label and pixel-level segmentation mask are related by $y^i = \mathbb{1}\left\{\sum_{k=1}^L S_k^i \geq 1\right\}$. Put simply, this means if the class label is positive, there is at least one pixel labeled as positive in the segmentation mask and vice versa. We train a segmentation network $f(\cdot)$ to produce a predicted segmentation mask $f(\boldsymbol{X}^i) = \hat{\boldsymbol{S}}^i$ by minimizing the cross entropy loss $CE(\hat{\boldsymbol{S}}^i, \boldsymbol{S}^i)$. For notation's sake, we will drop the sample index $i$, thus referring to the $i^{th}$ image $\boldsymbol{X}^i$ simply as $\boldsymbol{X}$ and the $k^{th}$ pixel $X_k^i$ simply as $X_k$.

**Data generating process.** To study segmentation vs. classification, we take an information theoretic approach and use the following simplified data generating process. First, the pixel-level segmentation labels $\{S_1, \ldots, S_L\}$ are drawn from a joint distribution $P_S$. Then, each pixel $X_k$ is drawn from the distribution $P_X(\cdot|S_k)$. As above, the image-level classification label $y$ can be defined in terms of the segmentation mask, $y = \mathbb{1}\left\{\sum_{k=1}^L S_k \geq 1\right\}$.

## 2.2 Analyzing segmentation vs. classification

To understand why segmentation and classification may perform differently on the same dataset, we consider the distributions that classification vs. segmentation networks are trained to discriminate. As an example, we'll use chest x-ray pneumothorax classification. In classification, supervision is applied at the image-level—the network seeks to discriminate images with a pneumothorax vs. images without a pneumothorax (Figure 2, left). In segmentation, supervision is applied at the pixel-level—the network seeks to identify pixels that contain a pneumothorax (Figure 2, right). Looking at these distributions, differences emerge.

**Sample complexity.** First, we see that segmentation has more annotated input-output pairs, as each pixel is annotated. Moreover, we see that classification's positive and negative classes share more features than segmentation's—implying segmentation may have greater class separability. We capture this notion of separability using the Kullback–Leibler (KL) divergence and show the following:

**Proposition 1** *It holds that*

$$D_{KL}\big(\Pr(X_k|S_k=0)||\Pr(X_k|S_k=1)\big) \geq \frac{\Pr(y=1)}{\Pr(S_k=1)} D_{KL}\big(\Pr(X_k|y=0)||\Pr(X_k|y=1)\big).$$

$$(1)$$

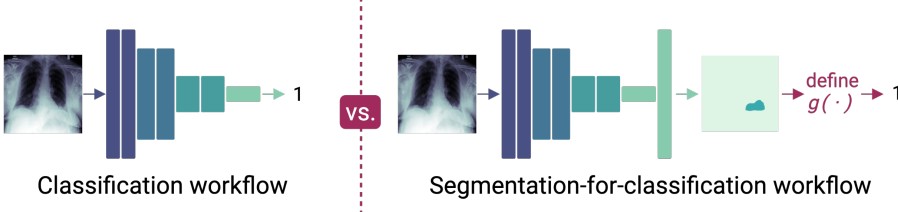

| Classification workflow | vs. | Segmentation-for-classification workflow |

Figure 3: Diagram showing how a chest x-ray disease classification problem can be framed using either standard classification or segmentation-for-classification. In the traditional classification workflow, the input image is processed by a classification network, which outputs a classification label vector. In the segmentation-for-classification workflow, an input image is processed by a segmentation network, which outputs binary masks showing where each abnormality is found. These masks are then converted into a classification label vector via a summarizing function, $g(\cdot)$.

In other words, the KL divergence between data given pixel-level labels (as is done in segmentation) is greater than or equal to the scaled KL divergence given image-level labels (as is done in classification). We note that the scaling factor $\frac{\Pr(y=1)}{\Pr(S_k=1)}$ is always greater than or equal to one. We provide the proof for Proposition 1 in Appendix A3.1, confirming our intuition that segmentation supervision results in more separable data distributions. Because of the greater quantity of input-output pairs and the greater KL divergence, we expect segmentation's discrimination function to be easier to learn, requiring fewer images or a simpler function class [22, 23]. Since we study the setting in which the segmentation mask directly indicates the classification label ($y = \mathbb{1}\left\{\sum_{k=1}^{L} S_k \geq 1\right\}$), we similarly expect segmentation-for-classification to have higher performance than standard classification in the limited data regime. Finally, from Proposition 1, we expect segmentation-for-classification to impart more benefit for tasks with small targets; for large targets, the terms in Proposition 1 approach one another.

**Robustness to spurious correlations.** The second observation we make from Figure 2 is that, because background features appear in classification's positive class, it is possible for the classification network to rely on background features to identify positive samples if those background features spuriously correlate with the class label. Such failure modes have been observed in prior work. For example, classifiers relying on metal L/R tokens to identify pneumonia [24]; classifiers looking for chest drains to classify pneumothorax [25]; and classifiers looking to image edges (which can change with patient positioning and radiographic projection) to classify COVID-19 [26]. Segmentation should be more robust to such background features, since background features are less correlated with the target class. This notion has been formalized in past work [21], which we recall here.

Specifically, for spurious features that do not overlap with the segmentation target, the mutual information between positive features and spurious features decreases as we transition from supervising with image-level labels to pixel-level labels. Since the spurious feature is less informative of the pixel-level labels than the classification label, we expect segmentation to be more robust to the presence or absence of spuriously correlated background features.

## 3 Methods

To use segmentation for classification tasks, we need to determine how to use segmentation networks to obtain classification labels. We'll do this by defining a *summarizing function*, $g(\cdot)$, which takes as input the image and segmentation network and outputs a class label $g(\boldsymbol{X}, f(\cdot)) = \hat{y}$. The segmentation-for-classification workflow is shown in Figure 3. In this section, we describe the summarizing functions that we evaluate. We note these methods can be used for both binary class problems and multiclass problems, as long as the classes of interest follow the task specification set above. We provide multiclass results in Appendix A5.6.

### 3.1 Rule-based summarizing functions

We first consider a simple, rule-based summarizing function. This method is intuitive: if there is a positive region in the predicted segmentation mask, we infer a positive classification label.

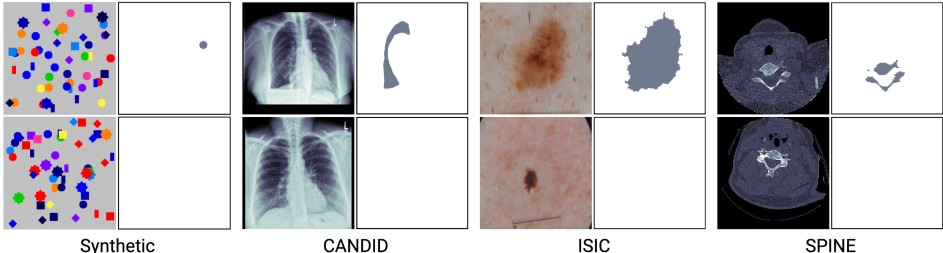

Synthetic     CANDID     ISIC     SPINE

Figure 4: Visualization of the datasets used in this manuscript: the synthetic dataset, where the task is to classify if a navy blue circle is preset; CANDID, a pneumothorax classification task; ISIC, a lesion classification task; and SPINE, a cervical fracture classification task. For each dataset, we show a positive and negative image and their corresponding segmentation masks.

Specifically, at inference, we binarize the probabilistic mask output from the segmentation network using threshold $t$. If the binarized mask contains over threshold $\tau$ total positive pixels, we return a positive classification label. To compute the class probability, we average the pixel-wise probabilities of all pixels in the class. Algorithms for this method are provided in Appendix A2.1.

## 3.2 Trained summarizing functions

Next we consider summarizing functions trained to transform the segmentation information into classification labels. The intuition behind these methods is that trained functions may learn more nuanced differences between true positive and true negative predicted segmentation masks. We train these summarizing functions using the same splits used to train the segmentation network, and freeze the segmentation networks before training the summarizing functions.

We first consider trained summarizing functions that operate on the predicted segmentation masks. We consider three architectures for the summarizing function, each with increasing complexity: a fully connected layer, a global average pooling layer followed by a fully connected layer, and existing image classification architectures such as SqueezeNet [27] and ResNet50 [28].

We also consider summarizing functions that operate on segmentation network embeddings, similar to traditional transfer learning. We consider both shallow and deep embeddings and two different summarizing function architectures: a simple architecture consisting of pooling and fully connected layers, and a more complex classification head proposed in past work [20].

We fully describe these summarizing functions and training processes in Appendix A2.2.

## 4 Experiments

In this section, we first confirm the takeaways from our analysis in Section 2 on a synthetic dataset (Section 4.1). Then, we evaluate the summarizing functions we described in Section 3 on medical imaging datasets (Section 4.2). We briefly describe the datasets and training below; we provide full details of the synthetic dataset and training in Appendix A4 and the medical datasets and training in Appendix A5.

**Datasets.** The synthetic dataset consists of many colored shapes on a gray background; the task is to determine if a navy blue circle is present. We also evaluate three medical imaging datasets: CANDID, in which we aim to classify pneumothorax in chest x-rays (n=19,237) [29]; ISIC, in which we aim to classify melanoma from skin lesion photographs (n=2,750) [30]; and SPINE, in which we aim to classify cervical spine fractures in CT scans (n=2,018, RSNA 2022 Cervical Spine Fracture Detection Challenge). These datasets are visualized in Figure 4. The SPINE dataset is an example of where the growing number of off-the-shelf networks enables segmentation-for-classification: we use a pretrained CT segmentation network to generate segmentation targets for this task, as detailed in Appendix A5.

**Training.** To establish baseline classification performance, we train a standard supervised classification network. To establish baseline segmentation+classification performance, we train a multitask learning network where the classification and segmentation networks are trained concurrently and

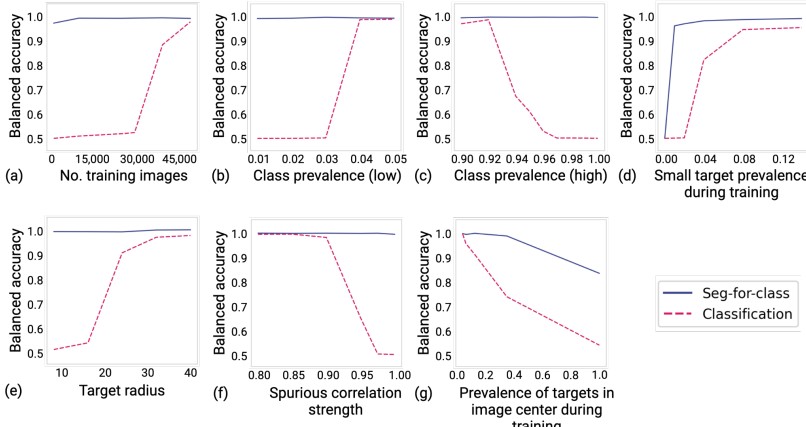

Figure 5: Results from experiments on the synthetic datasets. Each panel shows classification vs. segmentation-for-classification performance, as measured by the balanced accuracy, on the y-axis as a characteristic of the training dataset is changed on the x-axis.

share the same network backbone using the Y-Net structure [31]. To obtain segmentation-for-classification performance, we implement each summarizing function as described in Section 3. We train classification, multitask, and segmentation networks by minimizing the cross entropy loss between the predicted and ground truth labels using the Adam optimizer. We provide additional training details and information on hyperparameter tuning in Appendix A4.2 and A5.2, and we provide ablation experiments controlling for segmentation vs. classification model capacity in Appendix A5.3.

## 4.1   Experiments on synthetic data

We first perform experiments on synthetic data to confirm the takeaways from our analysis. By using a synthetic dataset, we have precise control over dataset and task characteristics, allowing us to isolate and adjust these variables to assess model performance. We provide highlights of our synthetic experiments below; a full description of the experimental procedure on synthetic data can be found in Appendix A4. In these synthetic experiments, we use the rule-based summarizing function.

The first takeaway from Section 2 is that segmentation-for-classification should require fewer images to achieve high performance, particularly for tasks with small targets. We examine this with our synthetic dataset in the following experiments.

- We sweep the number of training images from 1,000 to 100,000 and plot classification and segmentation-for-classification balanced accuracy in Figure 5a, where we see segmentation-for-classification has improved performance in the limited data regime.
- Similar to learning with small datasets, we also expect segmentation to perform better with high or low class prevalence, which we see in Figure 5b and Figure 5c.
- Segmentation's ability to learn with fewer examples should also extend to rare subtypes—we expect higher performance on class subtypes that appear infrequently during training. To evaluate this, we train on small, medium, and large navy blue circles and evaluate network performance on the small circles as they are made increasingly rare during training (Figure 5d). We see segmentation-for-classification achieve higher performance on this data slice with few training examples.
- We vary the object size in the train and test dataset and find segmentation-for-classification provides more boost with smaller targets (Figure 5e).

The second takeaway from our analysis is that segmentation-for-classification should be more robust to background features that spuriously correlate with the class of interest. We evaluate this with two synthetic experiments.

- We first evaluate an obvious case of spurious correlation: when a background object correlates with the class label. We make a large, pink square increasingly correlated with

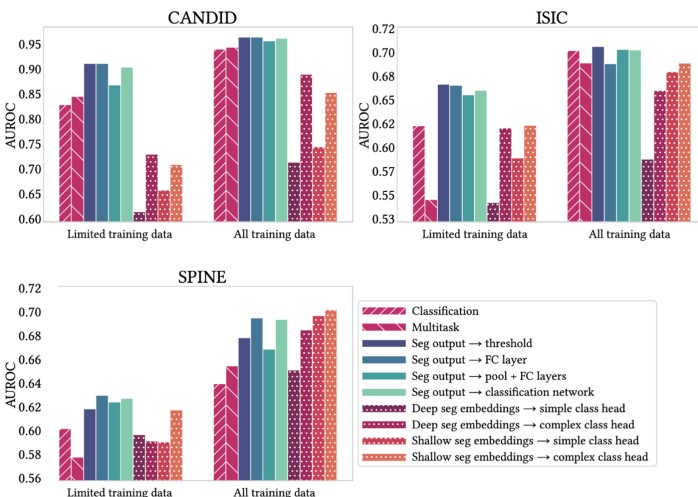

Figure 6: Comparison of different segmentation-for-classification methods on the CANDID, ISIC, ad SPINE datasets in both the limited and abundant training data settings. As baselines, we include classification and multitask network performance. FC: fully connected.

- the target (a small navy blue circle) in our synthetic dataset—then remove this correlation in the test dataset—and show that segmentation is robust to this correlation (Figure 5f).
- Spurious background correlations can also include less obvious features, like where the target object occurs; in other words, we expect segmentation to be more robust to target location. We explore this by changing where the target navy blue circle appears during training, but allowing the navy blue circle to appear anywhere in the test dataset. As the target is increasingly restricted to a particular location during training, classification is less able to generalize to finding the target in other regions of the image (Figure 5g).

## 4.2 Experiments on medical imaging data

Given the promising results on synthetic data, we next evaluate performance on the medical datasets and evaluate segmentation-for-classification training methods. In Section 4.2.1 we compare the performance of different summarizing functions, classification networks, and multitask networks. In Section 4.2.2 we evaluate semi-supervised segmentation-for-classification.

### 4.2.1 Comparing summarizing functions, classification, and multitask training

We evaluate each summarizing function and model on the three medical datasets in two regimes: limited labeled data, which consists of 10% of randomly sampled training data, and all labeled training data. We report mean AUROC on the test set in Figure 6. We observe that the simple, rule-based segmentation-for-classification method achieves higher mean AUROC than both the traditional classification and multitask networks for all tasks and amounts of training data. Specifically, we observe a 10.0%, 6.9%, and 2.8% improvement in mean test AUROC in the limited training data regime for CANDID, ISIC, and SPINE respectively over traditional classification, and a 2.6%, 0.6%, and 6.1% performance improvement in the all training data setting.

Comparing summarizing functions, we find that summarizing functions which operate over segmentation network outputs generally outperform those that operate over embeddings. Among summarizing functions that operate over segmentation outputs, three perform similarly well: thresholding the segmentation output, learning a fully connected layer, and training a full classification network on top of the segmentation output. The threshold-based summarizing function has additional advantages: it does not require training, thus saving training effort and avoiding potential overfitting to training data; it is fast to compute at inference; and the results are intuitive—when there is a segmentation mask, there is an associated positive class label. For these reasons, we recommend simple rule-based summarizing functions for converting a segmentation mask into a classification label. We compare the error modes of these methods in Section 5.

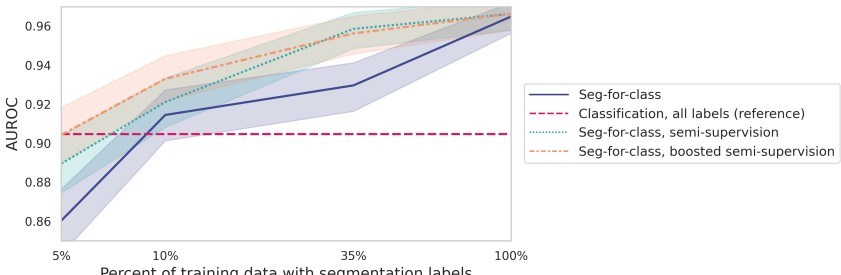

Figure 7: Semi-supervised segmentation-for-classification. As the x-axis increases, all models are trained with additional segmentation labels; the semi-supervised model is also trained with unlabeled data, and the boosted semi-supervised model is also trained with data that only has classification labels. Each line plots the average AUROC of the test set with shaded 95% confidence intervals.

### 4.2.2 Semi-supervised segmentation-for-classification

As discussed in the Introduction, a historical obstacle to widespread application of segmentation is the difficulty of curating labeled training datasets. However, recent work has led to more public datasets, pretrained networks, and methods to reduce the labeling burden of training segmentation models. We posit that this progress in segmentation will also benefit segmentation-for-classification. In the previous section, we showed that an off-the-shelf spine segmentation model could be used to generate segmentation labels for training the SPINE segmentation-for-classification models. In this section, we verify that a semi-supervised segmentation method can improve performance of segmentation-for-classification models. We consider two common settings:

- **Setting 1, additional unlabeled data.** Because unlabeled data is often abundant, practitioners may have a subset of their dataset labeled with segmentation masks and a larger set of unlabeled images.
- **Setting 2, additional classification data.** Other practitioners may have a subset of their dataset labeled with segmentation masks and a larger set of images with classification labels.

For Setting 1, we apply a previously-published semi-supervised segmentation method which takes a small set of labeled data and a large set of unlabeled data as input and trains a segmentation model using a combination of data augmentation, consistency regularization, and pseudo labeling [32].

For Setting 2, we update the previously-proposed methods by using the classification labels to improve the pseudo segmentation labels, which we call "boosted" semi-supervision. The original semi-supervised approach uses pseudo labeling, in which a small amount of labeled segmentation data is used to train an initial segmentation network; the initial network predicts pseudo segmentation masks for all unlabeled data; and a final network is trained on the pseudo segmentation masks. In the "boosted" algorithm, we use the available classification labels to modify the pseudo segmentation masks. Specifically, for images that have negative classification labels, we zero out the pseudo segmentation masks: because we know the classification label is negative, we know that there should not be any positive region in the segmentation mask. For images that have a positive classification label, we check to see if the pseudo segmentation mask has a positive segmentation region; if it does not have a positive segmentation region, we know that the pseudo segmentation mask is wrong and we exclude the sample from training of the final segmentation network.

These semi-supervised approaches learn from the unlabeled data to train a better segmentation model, which we hypothesize will also improve segmentation-for-classification performance. In these experiments, we use the threshold-based summarizing function and train models using a balanced training dataset. We report model performance for varying amounts of labeled training data in Figure 7. We report additional performance metrics and the number of labels for each method at the most limited and abundant data settings in Table A8.

From these results, we observe that all of the segmentation-for-classification models trained with 10% of the training data labeled exceed the performance of the classification model trained with 100% of the training data labeled. These results confirm one of the takeaways from our analysis, which is that segmentation-for-classification can achieve higher performance with fewer images. Additionally, we see that the semi-supervised training further improves segmentation-for-classification performance,

confirming that semi-supervised segmentation methods can directly be used for segmentation-for-classification. Finally, we see that the boosted semi-supervised approach indeed improves performance beyond standard semi-supervision in the limited data regime.

# 5 Benefits and drawbacks of segmentation-for-classification

In this section, we tie together our analysis, synthetics, and experiments on real data to summarize the benefits and drawbacks of using segmentation-for-classification.

**Segmentation can improve aggregate performance, particularly with small datasets.** Using segmentation-for-classification can lead to higher-performing models compared to traditional classification. Our analysis in Section 2 shows this benefit is due in part to greater divergence between segmentation's positive and negative classes and higher quantity of annotation (we evaluate the contributions of each of these factors with an additional experiment in Appendix A4.3). We observe this takeaway empirically in Figure 6, showing segmentation-for-classification improves mean performance on all datasets and training settings. Our analysis suggests greater performance differences occur in the limited data regime, which can include:

- **Small datasets.** We expect greater boosts from segmentation-for-classification with limited dataset sizes. We observe this with synthetics in Figure 5a and on two of the three medical datasets in Figure 6. Small datasets often occur when developing models on custom data or when training models for rare diseases.
- **Low class prevalence.** Segmentation can also achieve higher performance than classification when the class balance is very low or high (shown with synthetic datasets in Figure 5b,c), since segmentation is able to better learn from the few examples of the low-prevalence class.
- **Rare subtypes.** We expect segmentation-for-classification models to have improved ability to identify rare subtypes, which appear infrequently in the training dataset. We observe this on the synthetics (Figure 5d) and see this property revealed in the medical datasets as well: on small lesions that are positive for melanoma, which are less commonly seen during training, traditional classification achieves a recall of 0.18 and segmentation-for-classification achieves a recall of 0.47.

Finally, the analysis in Section 2 suggests segmentation provides greater benefit for tasks with small targets. We observe this with synthetics in Figure 5e. On the real data experiments, models trained with the full datasets reflect this property as well: the SPINE dataset has the lowest average target-to-background size ratio (with the target taking up an average 0.6% of the image) and achieves the greatest performance boost with segmentation-for-classification, while the ISIC dataset has the highest target-to-background size ratio (with the target taking up an average 26.3% of the image) and sees the lowest performance boost.

**Segmentation reduces susceptibility to spurious correlations.** A worrying failure mode of classification networks is that instead of learning to identify the pathology of interest, the model instead looks for easier-to-identify features that spuriously correlate with the true target—such as the examples given in Section 2. Segmentation-for-classification should be more robust to background features spuriously correlated with the target task.

Empirically, we observe this in our synthetic data (Figure 5f) and evaluate a spurious correlation on the medical datasets here. Specifically, in the CANDID dataset, we evaluate performance under a known spurious correlation condition: most pneumothoraces in chest x-ray datasets co-occur with a chest tube, which are used to treat pneumothorax [25]. Thus, it is important that the pneumothorax classification algorithms are able to find pneumothoraces without chest tubes, as those are the patients who have not yet been identified and treated. To evaluate each model's robustness to this spurious correlation, we evaluate model performance on patients who do not exhibit the spurious correlation (i.e., patients with pneumothorax but no chest tube or patients without pneumothorax but do have a chest tube) and observe that segmentation-for-classification achieves an AUROC of 0.84 while the classifier's AUROC drops to 0.58.

We also expect segmentation-for-classification to be more robust to the target location, which is another type of spurious correlation. We observe this in the synthetics (Figure 5g) and on the medical data: recall of the CANDID classifier drops up to 22.5% conditioned on the lung region in which the pneumothorax occurred (dropping from 0.92 recall in the right lower hemithorax to 0.71 recall on the

left upper hemithorax) while segmentation-for-classification recall drops at most 7.5% (from 0.94 in the left lower hemithorax to 0.87 in the right upper hemithorax).

**Segmentation facilitates human assessment.** Segmentation-for-classification inherently delivers location information. While there are methods to probe classification models for which areas of an image contribute to the model's prediction (e.g., saliency maps [33], class activation maps [34, 35]) these methods are often unreliable; shortcomings are further discussed in prior work [36, 37]. By providing location information as well as classification information, segmentation-for-classification models enable the user to more effectively adjudicate model findings. This location information is more pertinent for some applications where the abnormality may be hard to find (e.g., pneumothorax, fracture) versus use cases where the visual features used by the model are obvious (e.g., skin lesion).

**Segmentation enables more precise model evaluation.** We can perform more detailed model evaluation and error analysis using segmentation-for-classification's location and size information. For example, we can evaluate models with a stricter definition of recall: instead of identifying a true positive as when the image-level label is positive and correct, we can also require that the correct region of the image is identified. When we compute the stricter recall metric for models trained with all labeled data, we see reductions in recall: CANDID drops from 0.87 to 0.85 and SPINE drops from 0.74 to 0.55. We can't similarly assess the classification algorithms with this stricter definition of recall because of the lack of reliable methods to acquire accurate location information from classifiers. However, precise model evaluation is important for low-failure-tolerance applications such as medicine, and segmentation-for-classification provides an obvious means of doing so. Finally, we can perform additional error analysis using segmentation-for-classification's size and location information, such as assessing the average abnormality size and region.

**Segmentation has a higher per-image labeling cost.** The burden of generating segmentation training labels is a drawback of training segmentation networks and can change for different tasks (e.g., 3D images have a higher labeling burden). In comparison, classification is cheap to generate labels for—particularly for cases where classification labels can be pulled from electronic health records [38]. In this work, we've explored two ways of reducing segmentation's labeling burden while improving classification performance: using an off-the-shelf segmentation model and using semi-supervised training methods. As new tools emerge and make it easier to generate segmentation training labels [13, 14, 15], we expect the labeling burden gap between classification and segmentation to lessen. Still, the increased annotation cost of segmentation given available tools should be weighed against the expected benefits (detailed above) of using segmentation-for-classification on a given dataset.

## 6 Conclusion

Historically, segmentation networks have been employed only when required due to their high annotation cost. Given the increasing ease of training segmentation networks, we aim to better flesh out the decision space between training a segmentation vs. classification network. In Section 2, we develop intuition and formally analyze why segmentation and classification networks can perform differently. In Section 3 and 4, we explore methods to implement segmentation-for-classification and empirically evaluate segmentation-for-classification over many datasets and training conditions. Finally, in Section 5, we discuss the benefits and drawbacks of segmentation-for-classification. We show that segmentation-for-classification can lead to more performant, robust, and interpretable models. In clinical settings, one may prefer more these upsides despite the added annotation cost.

Broadly, these results underline the potential of deep learning-based segmentation as a general tool for automating image analysis. Segmentation data provides an information-rich environment for deep neural networks to learn from, which we've shown results in separable, robust embedding spaces. In turn, segmentation network outputs capture detailed information about images, which can support many downstream analysis tasks.

In future work, we are interested in how to leverage segmentation for targets that don't fit the task specification we study here, perhaps utilizing anatomical or coarser forms of segmentation to still reap some of the benefits of training segmentation models. Additionally, we are interested in reader studies focused on quantifying the value of the location information contained in segmentation masks for radiologists working with AI algorithms. Finally, we are interested in how to leverage segmentation to benefit other machine learning applications in radiology, extending our study of how task framing impacts a model's properties.

## Acknowledgements

We thank Dr. Leslie Smith and Dr. Darvin Yi for their discussion and ideas related to this topic.

Sarah Hooper is supported by the Fannie and John Hertz Foundation, the National Science Foundation Graduate Research Fellowship under Grant No. DGE-1656518, and as a Texas Instruments Fellow under the Stanford Graduate Fellowship in Science and Engineering. Additionally, we gratefully acknowledge the support of NIH under No. U54EB020405 (Mobilize), NSF under Nos. CCF1763315 (Beyond Sparsity), CCF1563078 (Volume to Velocity), and 1937301 (RTML); US DEVCOM ARL under No. W911NF-21-2-0251 (Interactive Human-AI Teaming); ONR under No. N000141712266 (Unifying Weak Supervision); ONR N00014-20-1-2480: Understanding and Applying Non-Euclidean Geometry in Machine Learning; N000142012275 (NEPTUNE); NXP, Xilinx, LETI-CEA, Intel, IBM, Microsoft, NEC, Toshiba, TSMC, ARM, Hitachi, BASF, Accenture, Ericsson, Qualcomm, Analog Devices, Google Cloud, Salesforce, Total, the HAI-GCP Cloud Credits for Research program, the Stanford Data Science Initiative (SDSI), and members of the Stanford DAWN project: Facebook, Google, and VMWare. Figures in this paper were created with BioRender. The U.S. Government is authorized to reproduce and distribute reprints for Governmental purposes notwithstanding any copyright notation thereon. Any opinions, findings, and conclusions or recommendations expressed in this material are those of the authors and do not necessarily reflect the views, policies, or endorsements, either expressed or implied, of NIH, ONR, or the U.S. Government.

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

# Appendices

## A1    Related work

**Segmentation to benefit classification.** Our study is inspired by past works that employ segmentation networks in classification problems and report various benefits of doing so. *Improved overall accuracy.* For example, [16] uses a segmentation network to generate segmentation masks of different tissue types in lung ultrasound images, then trains a sequence of global average pooling and fully connected layers to convert the tissue masks into classification labels. The authors report higher aggregate performance compared to training a classification model from scratch for their lung ultrasound classification task. In [17], the authors expand this approach by including diagnostic tissue segmentation targets (i.e., segmenting benign vs. malignant tissue as different segmentation classes), then converting the segmentation masks into class labels via a likelihood score over the segmented tissue types. Similarly, [18] jointly segments and classifies liver lesions in CT scans by segmenting the lesion as one of multiple possible lesion types and finds that the segmentation approach results in higher classification performance than traditional classification or multitask learning. *Training with fewer labels.* Other reports focus on segmentation networks' ability to learn from limited labeled data. For instance, [19] converts segmentation outputs into classification labels via a likelihood score and compares the aggregate performance of the segmentation network against classification and multitask learning networks when trained with varying dataset sizes. The authors find that the segmentation network achieves higher classification scores in the limited data regime. Similarly, [20] uses features from a UNet pretrained on a segmentation task to fine tune a classifier and again finds higher classification performance using the pretrained segmentation weights in the limited data regime. *Improved robustness.* Finally, [21] shows that training a segmentation network instead of a classification network results in predictors more robust to spurious correlations.

**Multitask learning.** Other works have combined segmentation and classification via multitask learning, in which the classification and segmentation networks share the same backbone but different task heads trained concurrently [39, 40, 41, 31, 42]. The multitask approach differs from the approaches discussed above, which each train a segmentation network, freeze it, then use the segmentation network's embeddings or outputs to produce a classification label. In multitask frameworks, the classification task can impact the weights learned for the segmentation task via the shared backbone, and training schedules and task weighting must be managed.

**Classification to benefit segmentation.** Other studies focus on the reverse of what we are interested in by first training a classification network then using that classifier to generate segmentation information. These approaches typically aim either to generate segmentation labels for "free" by extracting location information from the classification models or to use this location information to improve a classification network's interpretability. Examples of this class of related work include using saliency maps [33], class activation maps [34, 35], occlusion maps [43], and classifiers with built-in attention mechanisms [44].

**Semantic segmentation.** A great deal of work has focused on improving semantic segmentation methods [45, 46], which underlie the segmentation-for-classification methods we explore here. Segmentation-for-classification frameworks contain an additional step converting a semantic segmentation network or output into an image-level classification label, particularly for tasks which are not typically framed as segmentation tasks. Additionally, many recent works have advanced our ability to train semantic segmentation networks with limited labeled data, including using self-supervised networks, semi-supervision, and foundation models [4, 5, 6, 7, 47, 8], many of which have been adapted specifically to medical imaging [48, 49, 50]. These methods make training segmentation-for-classification models less labor-intensive.

**Transfer learning.** The segmentation-for-classification framing can be viewed as a type of transfer learning. While many transfer learning studies have focused on settings where the task type stays consistent (e.g., transferring one classification network to a different classification task), some studies have looked at transferring classification weights to segmentation tasks or vice versa [51, 52, 53]. Some segmentation-for-classification methods can be considered transfer learning, where

---
**Algorithm 1** Compute binary label
---
**Require:** $ProbSegMask \in [0,1]^{H \times W \times D}, t \in [0,1], \tau \in \mathbb{Z}^*$.
 1: $BinSegMask \leftarrow ProbSegMask \geq t$
 2: **if** $sum(BinSegMask) \geq \tau$ **then**
 3:     **return** 1
 4: **else**
 5:     **return** 0
 6: **end if**
---

the segmentation network is trained first and much of the segmentation network is frozen before transferring the learned weights to a classification task.

**Label granularity.**  A related line of work explores the right label granularity for training classification models (e.g., training "herbivore vs. carnivore" compared to training "horse vs. deer vs. cat vs. dog vs...") [54, 55]. Similar to our findings, these works find that more granular labels can lead to improved network optimization and generalization [54].

**Modified classification networks.**  Finally, many works have modified straightforward supervised classification training schemes. For example, self- and semi-supervised learning aim to improve performance with limited labeled data [56], and past work has proposed various modifications to classification training to improve robustness [57, 58].

In this manuscript, we use straightforward classification as we are interested in exploring fundamental differences between classification and segmentation. We observed that many performance benefits can be achieved simply by using a segmentation model, improving performance along many axes without specialized tooling. Conversely, classification modifications are typically different for different objectives (e.g., methods for reducing reliance on spurious correlations are different from methods that handle class imbalance are different from methods that improve interpretability). We believe the fact that segmentation results in many benefits without specialized tooling for each is a major strength of the proposed approach.

**Our contributions.**  We first analyze why we see performance differences, supporting and helping to explain the empirical results that we and others have observed as well as the conditions under which we can expect to see benefits. Next, we implement both previously proposed and new methods for using segmentation-for-classification and compare these approaches across multiple datasets using varying amounts of training data. We perform this head-to-head comparison to build up a recommendation of best practices for using segmentation networks in classification problems, including: recommending a summarizing function; showing an off-the-shelf segmentor can achieve improved classification performance; showing an existing semi-supervised training method can improve performance; and adapting that semi-supervised method specifically to the segmentation-for-classification setting. Finally, we aggregate and expand on the benefits of switching from classification to segmentation. We expand on benefits by showing improved performance with low class prevalence, rare subtypes, and location spurious correlations, and we discuss these quantitative benefits in conjunction with qualitative benefits like improved interpretability and model evaluation. We use this consolidated evidence to make the broader case that many classification problems would benefit from a segmentation framing.

## A2   Summarizing functions

Below we provide additional details on the summarizing functions we evaluate in the main text.

### A2.1   Rule-based summarizing functions

In Section 3.1, we introduce a rule-based summarizing function using thresholds to transform the segmentation output into a classification label. Here we give further details of that summarizing function. Specifically, in Algorithm 1, we give the algorithm we use to compute a binary, image-level label from a probabilistic segmentation mask. Given a probabilistic segmentation mask—which

**Algorithm 2** Compute probabilistic label

---

**Require:** $ProbSegMask \in [0,1]^{H \times W \times D}, t \in [0,1], BinaryLabel \in \{0,1\}.$
1: **if** $BinaryLabel = 1$ **then**
2:     $PosProbs \leftarrow ProbSegMask[ProbSegMask \geq t]$
3:     **return** $mean(PosProbs)$
4: **else**
5:     $NegProbs \leftarrow ProbSegMask[ProbSegMask < t]$
6:     **return** $mean(NegProbs)$
7: **end if**

---

contains the segmentation network output, the pixelwise probabilities indicating where the class of interest was found—we first binarize the segmentation mask using threshold $t$. At default, $t = 0.5$. We then sum all pixel values in the binary segmentation mask; if there are more than threshold $\tau$ positive pixels in the segmentation mask, we return a positive binary image-level label. In this work, we set $\tau = 100$.

In Algorithm 2 we provide the method we use to compute a probabilistic image-level label from a probabilistic segmentation mask. We use these probabilistic labels to compute the AUROC. Given a probabilistic segmentation mask, we first compute the binary label using Algorithm 1. If the binary label is positive, we compute the probabilistic image-level label as the mean of pixel values that are greater than threshold $t$. Similarly, if the binary label is negative, the probabilistic image-level label is the mean of pixel values less than $t$. At default, $t$ takes value 0.5.

### A2.2   Trained summarizing functions

In Section 3.2, we introduce trained summarizing functions to transform the segmentation output into a classification label. We give further details of those summarizing functions here.

**Summarizing functions that operate over segmentation outputs.**   After first training the segmentation network, we generate predicted segmentation masks $\hat{S}$ for all training and validation images. Note that we do not binarize these masks, but retain the pixel-wise probabilities. Then, using the same dataset splits used for training the segmentation network, we train the summarizing function $g(\cdot)$ to take the predicted segmentation mask $\hat{S}$ as input and produce a class label vector $g(\hat{S}) = \hat{y}$ by minimizing the cross entropy loss $CE(\hat{y}, y)$. At inference, the segmentation network first generates a predicted segmentation mask for the test image, then the predicted mask is fed into the summarizing function which produces class probabilities.

We consider three architectures for trained summarizing functions that operate over segmentation network outputs. The first and simplest architecture is a fully connected layer on the flattened predicted segmentation mask $\hat{S}$, which is used to produce class scores that are normalized with a softmax function.

The next summarizing function we consider is a global average pooling layer followed by a fully connected layer. We take the output segmentation mask $\hat{S}$ and produce a vector with values representing the average of each class's segmentation mask. The fully connected layer then transforms these class averages into scores for each class, which are normalized with a softmax function. This method is similar in spirit to thresholding as described above but with a threshold learned from the values observed in the train set. This global average pooling to fully connected approach was previously proposed in [16].

Finally, we consider known image classification architectures as summarizing functions. At a high level, the task of transforming a segmentation mask into a classification label is exactly an image classification task: does the "image" represented by the predicted segmentation mask correspond to a positive or negative class label? So, we investigate if existing image classification architectures are suited for learning mappings from predicted segmentation masks to class labels. In this work, we consider SqueezeNet [27] and ResNet50 [28] as summarizing functions on top of the segmentation output.

| Model | Deep embedding layer name | Shallow embedding layer name |
|---|---|---|
| `models.segmentation.fcn_resnet50` | `backbone.layer2` | `backbone.layer4` |
| `Generic_UNet` [59] | `conv_blocks_localization[2]` | `conv_blocks_localization[4]` |

Table A1: Layer names we pull embeddings from for the trained summarizing functions. These names are from public networks available in Pytorch.

**Summarizing functions on pretrained segmentation embeddings.** We next consider summarizing functions that operate over segmentation network embeddings. We first train the segmentation network $f(\cdot)$, then store embeddings for all training and validation images. We train the summarizing function $g(\cdot)$ to take the segmentation embeddings as input and produce a class label vector $\hat{y}$ as output by minimizing the cross entropy loss $CE(\hat{y}, y)$. At inference, the image is passed through the pretrained segmentation network and embeddings from the segmentation network are stored. These embeddings are then processed with the summarizing function to produce class probabilities.

We consider two types of embeddings, shallow and deep. The intuition for exploring different embedding depths is that different depths contain different information about the input image, which may vary in their usefulness for the classification task. We give the name of the layer in the Pytorch model that we pull embeddings from in Table A1. We also consider two different architectures to classify the segmentation network's embeddings. First, we define a simple network architecture consisting of a global average pooling layer followed by three fully connected layers, which produces class scores that we normalize with a softmax function. Second, we consider a more complex classification head proposed previously in [20]. The classification head takes the segmentation embeddings as input, then processes the embeddings with three blocks of convolution-batch normalization-ReLU-maximum pooling operations. The convolutional kernels are size 3x3 and each produce 16 channels; the maximum pooling operation operates over a 2x2 window. The resulting embedding is flattened. Then, we process the flattened embedding with two blocks of linear transform-batch normalization-ReLU-dropout. The linear layer produces 100 features and the dropout layer has a probability of dropout set to 0.25. Finally, the resulting feature map is processed with a single linear layer to produce the final output, a vector equal in length to the number of classes, which can be normalized with the softmax function.

## A3  Supporting analysis

### A3.1  Proof of Proposition 1

*Proof.* We can write $\Pr(X_k|y = 1)$ as

$$\Pr(X_k|\exists\, j\, s.t.\, S_j = 1) = \frac{\Pr(\exists\, j\, s.t.\, S_j = 1, X_k)}{\Pr(\exists\, j\, s.t.\, S_j = 1)} = \frac{\sum_{S_1,\ldots,S_L:\sum S_k \geq 1} \Pr(S_1,\ldots,S_L)\Pr(X_k|S_1,\ldots,S_L)}{\Pr(\exists\, j\, s.t.\, S_j = 1)} \tag{A1}$$

$$= \frac{\sum_{S_1,\ldots,S_L:\sum S_k \geq 1} \Pr(S_1,\ldots,S_L)\Pr(X_k|S_k)}{\Pr(\exists\, j\, s.t.\, S_j = 1)} \tag{A2}$$

$$= \frac{\sum_{S_{-k}} \Pr(S_k = 1, S_{-k})\Pr(X_k|S_k = 1) + \sum_{S_{-k}:\sum S_{-k} \geq 1} \Pr(S_k = 0, S_{-k})\Pr(X_k|S_k = 0)}{\Pr(\exists\, j\, s.t.\, S_j = 1)} \tag{A3}$$

$$= \frac{\Pr(S_k = 1)\Pr(X_k|S_k = 1) + \Pr(X_k|S_k = 0)\sum_{S_{-k}:\sum S_{-k} \geq 1} \Pr(S_k = 0, S_{-k})}{\Pr(\exists\, j\, s.t.\, S_j = 1)} \tag{A4}$$

$$= \lambda \Pr(X_k|S_k = 1) + (1 - \lambda)\Pr(X_k|S_k = 0), \tag{A5}$$

where $\lambda = \frac{\Pr(S_k=1)}{\Pr(\exists\, j\, s.t.\, S_j=1)}$. Therefore, we have written the probability of a pixel value given the image-level label, $\Pr(X_k|y = 1)$, as a mixture of the probability of the pixel value given the pixel-level label, $\Pr(X_k|S_k = 1)$ and $\Pr(X_k|S_k = 0)$. Next, we can write $\Pr(X_k|y = 0)$ as $\Pr(X_k|S_1,\ldots,S_L = 0) = \Pr(X_k|S_k = 0)$ by our data generating process.

Putting together $\Pr(X_k|y = 1)$ and $\Pr(X_k|y = 0)$, the right hand side of the inequality in Proposition 1 is

$$D_{\mathrm{KL}}(\Pr(X_k|y = 0)\,||\,\Pr(X_k|y = 1)) = D_{\mathrm{KL}}(\Pr(X_k|S_k = 0)\,||\,\lambda \Pr(X_k|S_k = 1) + (1 - \lambda)\Pr(X_k|S_k = 0)) \tag{A6}$$

$$\leq \lambda D_{\mathrm{KL}}(\Pr(X_k|S_k = 0)\,||\,\Pr(X_k|S_k = 1)) + (1 - \lambda)D_{\mathrm{KL}}(\Pr(X_k|S_k = 0)\,||\,\Pr(X_k|S_k = 0)) \tag{A7}$$

$$= \lambda D_{\mathrm{KL}}(\Pr(X_k|S_k = 0)\,||\,\Pr(X_k|S_k = 1)) \leq D_{\mathrm{KL}}(\Pr(X_k|S_k = 0)\,||\,\Pr(X_k|S_k = 1)) \tag{A8}$$

where we have obtained the desired inequality due to convexity of the KL divergence. Note that by keeping $\lambda$ in the inequality, we can make the stronger claim

$$D_{\mathrm{KL}}(\Pr(X_k|S_k = 0)\,||\,\Pr(X_k|S_k = 1)) \geq \frac{\Pr(y = 1)}{\Pr(S_k = 1)} D_{\mathrm{KL}}(\Pr(X_k|y = 0)\,||\,\Pr(X_k|y = 1)). \tag{A9}$$

∎

| Variable name | Description | Default value |
|---|---|---|
| $n_{obj}$ | Number of total objects in the image | 50 |
| $n_{train}$ | Number of training images | $100,000$ |
| $n_{val}$ | Number of validation images | $20,000$ |
| $n_{test}$ | Number of test images | $100,000$ |
| $c_{target}$ | Color of the target object | navy blue |
| $r_{target}$ | Radius of the target object | 8 |
| $s_{target}$ | Shape of the target object | circle |
| $l_{target_x}$ | Possible x coordinates for the target object | $\sim unif\{0, 224\}$ |
| $l_{target_y}$ | Possible y coordinates for the target object | $\sim unif\{0, 224\}$ |
| $c_{spurious}$ | Color of the spurious object | pink |
| $r_{spurious}$ | Radius of the spurious object | 8 |
| $s_{spurious}$ | Shape of the spurious object | square |
| $l_{spurious_x}$ | Possible x coordinates for the spurious object | $\sim unif\{0, 224\}$ |
| $l_{spurious_y}$ | Possible y coordinates for the spurious object | $\sim unif\{0, 224\}$ |
| $c_{background}$ | Colors of the background objects | $\in$ {yellow, green, pink, red, orange, purple, eight shades of blue} |
| $r_{background}$ | Radii of the background objects | 8 |
| $s_{background}$ | Shapes of the background objects | $\in$ {circle, rectangle, diamond, octagon, star, square} |
| $l_{background_x}$ | Possible x coordinates for the background object | $\sim unif\{0, 224\}$ |
| $l_{background_y}$ | Possible y coordinates for the background object | $\sim unif\{0, 224\}$ |
| $p_{pos}$ | Probability of a positive class (i.e., class balance). At $p_{pos} = 0$ the dataset contains no positive samples; at $p_{pos} = 1$ the dataset contains only positive samples. | 0.5 |
| $p_{spur}$ | Strength of spurious correlation. At $p_{spur} = 1$ the spurious object always and only co-occurs with the target object; at $p_{spur} = 0$ the spurious object never co-occurs with the target object, but appears every time the target object is absent; at $p_{spur} = 0.5$, the spurious object has no dependence on the target object. | 0.5 |

Table A2: Variables that define a synthetic dataset. All experiments are run with datasets constructed from the default values unless specified otherwise.

| Experiment | Adjusted variable |
|---|---|
| Number of training images | $n_{train}$ is varied from 1,000 to 100,000. |
| Class balance | Class balance is changed in the train data by varying $p_{pos} \in [0.01, 1]$. Note that class balance in the validation and test data is not changed from default. |
| Size bias | Instead of all target objects having the same radius, we draw the radius from a distribution increasingly biased towards larger radii. Note that the validation and test datasets only show small objects. |
| Spurious correlation | Spurious correlation strength is changed in the train data by varying $p_{spur} \in [0, 1]$ and the spurious feature size is changed to 28. Note that spurious correlation strength in the validation and test data is not changed from default. |
| Location bias | The location of the target object is changed in the train data by varying $l_{target_x}$ and $l_{target_y}$ from $\sim unif\{112, 112\}$ to $\sim unif\{0, 224\}$. Note that location distributions are not changed from default in the validation or test data. |
| Target size | The size of the target and background objects are swept from 8 to 40. We change $n_{train}$ to 10,000. |
| Number of objects | The difficulty of the classification task is changed by sweeping the number of unique background objects from 1 to 10; the total number of objects $n_{obj}$ remains set to 50. We change $n_{train}$ to 25,000. |

Table A3: Experiments we run on synthetic datasets. For each experiment, we adjust the variable values indicated above (leaving the other variables at default) and assess how classification and segmentation-for-classification performances are impacted.

## A4 Synthetic dataset and experiments

We conduct experiments on a synthetic dataset to evaluate how dataset and task characteristics impact the performance of classification and segmentation-for-classification models. Here we provide additional details on our synthetic dataset and experiments. The synthetic dataset is visualized in Figure 4.

### A4.1 Synthetic dataset

The synthetic dataset is made up of images containing a gray background and many objects of varying shapes and colors. The task is to classify if a given image has a certain colored shape present. Each synthetic dataset is defined by the set of parameters given in Table A2.

To construct a synthetic image, we initialize a gray background of size (224, 224, 3). We then draw the class label $y \sim Bernoulli(p_{pos})$. We sample the spurious label $s$ according to the strength of the spurious correlation $p_{spur}$. Specifically, we define and sample the random variable $match \sim Bernoulli(p_{spur})$. If $match = 1$, we set the spurious label $s = y$; else, we set $s = 1 - y$. If $s = 1$, we create the spurious object specified by $(c_{spurious}, r_{spurious}, s_{spurious})$—the object's color, radius, and shape—and place the spurious object at the coordinates drawn from the distributions $(l_{spurious_x}, l_{spurious_y})$. We follow similar processes to generate and place the $n_{obj} - y - s$ background objects and target object. We place the target object last to ensure it is placed in the foreground.

To run experiments on the synthetic dataset, we vary one dataset variable, train the classification and segmentation-for-classification models, and evaluate performance. The variable we adjust for each experiment is described in Table A3. Additionally, we visualize the training dataset for each experiment in Figure A1.

### A4.2 Training details for synthetic datasets

We train a ResNet50 to classify the synthetic data and a ResNet50 with a fully convolutional segmentation head to segment the synthetic data. We train the classifier and segmentation networks by minimizing the cross entropy loss between the predicted and ground truth labels. We do not use data augmentation on the synthetic data. We train each network with an Adam optimizer and a learning rate of 1e-4, tuned from [1e-6, 1e-5, 1e-4, 1e-3]. We checkpoint the networks on the maximum balanced accuracy on the validation set. We use the threshold-based summarizing function.

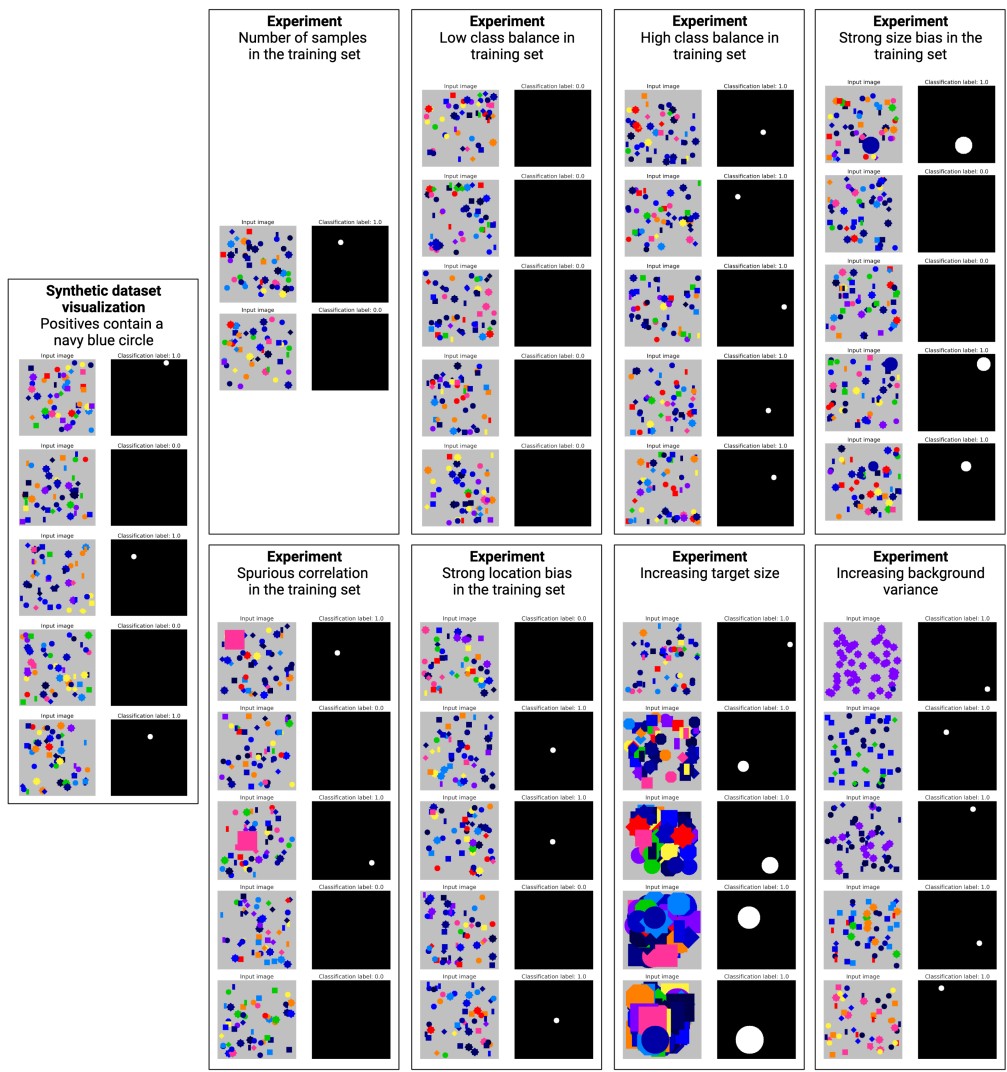

Figure A1: Visualization of the synthetic dataset task and experiments. The task is to identify if the image contains a navy blue circle. Each experiment changes a characteristic of the training dataset to observe how classification vs. segmentation-for-classification performance is impacted.

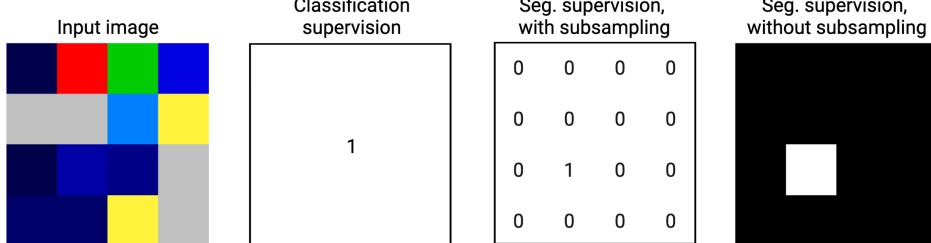

Figure A2: Visualization of the annotation quantity experiment. The input image is a grid of colored squares; classification supervision receives a class label per image; segmentation supervision with subsampling receives one class label per colored square; segmentation supervision without subsampling receives a class label per pixel.

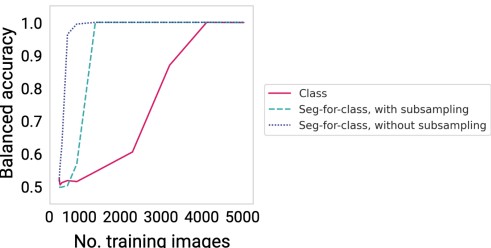

Figure A3: Results from annotation quantity experiment. This graph shows that providing segmentation supervision for every pixel is the most sample efficient, though providing supervision for every colored square still improves sample efficiency compared to providing one label for the entire image. This graph supports that it is both annotation quantity and the differences in class distribution leading to segmentation-for-classification's improved performance in the limited data regime.

### A4.3 Additional synthetic experiments

In the main text, we focus on the divergence between positive and negative classes as to why classification and segmentation perform differently. We do not explicitly model the benefit from denser annotation during training, instead taking it as a given that the denser annotation should help segmentation learn with fewer images. To show that annotation quantity is not the only component contributing to segmentation's improved performance, we run the following experiment.

We study a simplified setting visualized in Figure A2. Specifically, we divide a $112 \times 112$ grid into sixteen isotropic squares. The task is to identify if any of the squares are navy blue. Class balance is set to 0.5, meaning an image is as likely to have the target class as not. The remaining square colors are drawn randomly. In this simplified setting, we still expect divergence between segmentation classes to be greater than divergence between classification classes, following our analysis in Section 2.

Classification supervision is applied at the image-level, per usual. To understand the influence of annotation quantity, we evaluate segmentation-for-classification performance in two settings. First, we provide segmentation supervision with only one pixel per square. This level of supervision results in 16 annotated pixels per image, but an equal number of positive class annotations as the classification supervision. Second, we provide segmentation supervision over all pixels (this is the usual segmentation-for-classification setting), in effect providing 12,544 annotated pixels per image and $784\times$ as many positive class annotations. The supervision for each of these models is also visualized in Figure A2.

In effect, this experiment separates the impact of just divergence vs. divergence+higher quantity of (positive) annotations. We sweep the number of training images and report results in Figure A3.

We see that segmentation without subsampling outperforms segmentation with subsampling outperforms classification. This result confirms that both the denser annotation and greater divergence during training benefit segmentation-for-classification performance.

A greater number of annotated samples should improve most ML tasks, as has been extensively studied before. We focus on characterising the divergence as it is specific to segmentation-for-classification and less studied in prior work. Future work may study the relative benefits of these two aspects of supervision.

## A5   Real datasets and experiments

We run experiments on three public medical imaging datasets, described below and fully detailed in their original studies. The medical datasets are visualized in Figure 4.

### A5.1   Medical imaging datasets

CANDID: classifying pneumothorax in chest x-rays [29]

- **Classification task.** Our aim is to classify whether a chest x-ray contains a pneumothorax; ground truth classification labels are provided with the original dataset.
- **Segmentation target.** Our aim is to segment pneumothorax when it appears in a chest x-ray; ground truth segmentation masks are provided with the original dataset.
- **Dataset.** This dataset contains 19,237 chest x-rays. The images are two-dimensional, one-channel images of size 1024 x 1024, and we normalized each image to the range [0, 1]. We split this dataset randomly into 60% training images, 20% validation images, and 20% test images. In this dataset, 16.6% of images are positive for pneumothorax.

ISIC: classifying melanoma from photographs of skin lesions [30]

- **Classification task.** Our aim is to classify whether a skin lesion is melanoma or not; ground truth classification labels are provided with the original dataset.
- **Segmentation target.** Our aim is to segment the skin lesion when it is melanoma; ground truth segmentation masks are provided with the original dataset.
- **Dataset.** This dataset contains 2750 images of skin lesions. The images are two-dimensional, three-channel images that we resized to 224 x 224. We use the splits provided by the ISIC challenge organizers, resulting in 2000 training images, 150 validation images, and 600 test images. 18.9% of images are positive for melanoma.

SPINE: classifying cervical spine fractures in CT scans (RSNA 2022 Cervical Spine Fracture Detection Challenge)

- **Classification task.** Our aim is to classify whether there is a fracture in the cervical spine; ground truth classification labels are provided with the original dataset.
- **Segmentation target.** Our aim is to segment the vertabrae in which the fracture occurs. Unlike the previous two datasets, the ground truth segmentation masks were not available in this dataset for all training images. Instead, we used an off-the-shelf CT segmentation network [11] to generate segmentation masks of the vertebra. For each positive classification label, we create a corresponding segmentation mask which delineates the vertebrea in which the fracture occurs. This use case shows how off-the-shelf models can be used to convert some classification problems into segmentation problems without any additional labeling.
- **Dataset.** This dataset contains 2018 CT scans of the spine. The images are three-dimensional volumes with one channel. We applied a CT bone window then normalized each scan to the range [0, 1]. We resized each axial slice to 128 x 128 voxels using bilinear interpolation and resampled the axial dimension to 128 slices around the spine, resulting in 128 x 128 x 128 image volumes. 47.6% of these scans contain a cervical spine fracture. We randomly split this dataset into 60% training, 20% validation, and 20% test.

| Model | Dataset | Architecture | Learning rate | Weight decay |
|---|---|---|---|---|
| Segmentation | CANDID | **ResNet50**, SEResNet50, DeepLabv3, UNet | 1e-6, 1e-5, **1e-4**, 1e-3 | 1e-6, 1e-4, 1e-2, **0** |
| Segmentation | SPINE | ResNet50 3d, UNet 3d, **NNuNet 3d** | 1e-6, 1e-5, 1e-4, **1e-3** | 1e-6, 1e-4, 1e-2, **0** |
| Segmentation | ISIC | **ResNet50**, SEResNet50, DeepLabv3, UNet | 1e-6, 1e-5, 1e-4, **1e-3** | 1e-6, 1e-4, 1e-2, **0** |
| Classification | CANDID | ResNet50, SEResNet50, **DenseNet121**, ConvNext | 1e-6, 1e-5, **1e-4**, 1e-3 | 1e-6, 1e-4, 1e-2, **0** |
| Classification | SPINE | ResNet50 3d, UNet 3d encoder, **NNuNet 3d encoder** | 1e-6, 1e-5, 1e-4, **1e-3** | 1e-6, 1e-4, 1e-2, **0** |
| Classification | ISIC | ResNet50, SEResNet50, **DenseNet121**, ConvNext | 1e-6, **1e-5**, 1e-4, 1e-3 | 1e-6, 1e-4, 1e-2, 0 |
| Multitask | CANDID | **ResNet50** | 1e-6, 1e-5, **1e-4**, 1e-3 | 1e-6, 1e-4, 1e-2, **0** |
| Multitask | SPINE | **ResNet50 3d** | 1e-6, 1e-5, 1e-4, **1e-3** | 1e-6, 1e-4, 1e-2, **0** |
| Multitask | ISIC | **ResNet50** | 1e-6, 1e-5, **1e-4**, 1e-3 | 1e-6, 1e-4, 1e-2, **0** |

Table A4: Hyperparameters tested for the backbone networks; chosen hyperparameters are in bold.

| Summarizing function | CANDID (low data) | SPINE (low data) | ISIC (low data) | CANDID (high data) | SPINE (high data) | ISIC (high data) |
|---|---|---|---|---|---|---|
| Seg. output → fully connected layer | 1e-5 | 1e-6 | 1e-3 | 1e-5 | 1e-6 | 1e-4 |
| Seg. output → avg. pool + fully connected layer | 1e-2 | 1e-2 | 1e-3 | 1e-2 | 1e-2 | 1e-4 |
| Seg. output → SqueezeNet | 1e-5 | 1e-5 | 1e-5 | 1e-5 | 1e-6 | 1e-6 |
| Deep seg. embeddings → simple head | 1e-3 | 1e-3 | 1e-3 | 1e-3 | 1e-2 | 1e-3 |
| Shallow seg. embeddings → simple head | 1e-3 | 1e-4 | 1e-4 | 1e-4 | 1e-2 | 1e-3 |
| Deep seg. embeddings → complex head | 1e-4 | 1e-2 | 1e-3 | 1e-3 | 1e-4 | 1e-2 |
| Shallow seg. embeddings → complex head | 1e-3 | 1e-3 | 1e-4 | 1e-3 | 1e-3 | 1e-2 |

Table A5: Hyperparameters selected for training the summarizing functions.

## A5.2 Training details for medical imaging datasets

In Table A4 we list each hyperparameter we evaluated for the backbone networks trained on the medical datasets. We chose hyperparameters that maximized the validation AUROC. Chosen hyperparameters are in bold.

For the trained summarizing functions, we only tuned the learning rate. We evaluated learning rates [1e-6, 1e-5, 1e-4, 1e-3, 1e-2] for each summarizing function and chose the learning rate that maximized the validation AUROC. In Table A5 we list the chosen learning rate for each summarizing function and dataset.

To evaluate performance, we report mean AUROC over the test split of each dataset. We binarize the probabilistic labels to report additional performance metrics by setting the operating point on the ROC curve that maximizes the Younden Index. We compute 95% confidence intervals using bootstrapping (1000 iterations). We conduct our experiments using Pytorch Lightning (Pytorch version 1.9.0, Lightning version 1.5.10) [60, 61].

## A5.3 Ablations on model capacity

To report results in the main text, we tested multiple architectures for each task and dataset and selected the best-performing architecture for both segmentation and classification (Table A4). We did this to give each task and dataset the best performance, as it's not clear the best architecture for segmentation on a given dataset is the same that is best for classification.

However, segmentation networks are often higher capacity than classification networks due to their dense, pixel-wise outputs and corresponding decoders. To show that it is not simply a difference in model capacity that is leading to the observed performance differences between segmentation and classification, we perform the following ablations.

First, we use the same backbone for both segmentation and classification (Table A6). We use Resnet50 as the 2D backbone as it is a common backbone known to be useful for classification and segmentation. The Resnet50 classification head is a linear layer, while the Resnet50 segmentation head is the head described in the Fully Convolutional Network paper [62]. We use the UNet from the nnU-Net paper [59] for the 3D backbone. We see that segmentation shows the same or greater performance enhancement than what appeared in the original paper when using these standardized backbones. We further note that all of our synthetic experiments were run with a standardized backbone between classification and segmentation.

However, even when using the same backbone, segmentation models still typically have more parameters due to the convolutional decoder head. We next provide results comparing performance using

|  | Backbone | Method | AUROC |
|---|---|---|---|
| CANDID | ResNet50 | Classification | 0.80 |
|  |  | Seg-for-class | 0.91 |
| ISIC | ResNet50 | Classification | 0.56 |
|  |  | Seg-for-class | 0.67 |
| SPINE | nnU-Net | Classification | 0.60 |
|  |  | Seg-for-class | 0.62 |

Table A6: Performance comparison when classification and segmentation models are trained with the same network backbone. For the SPINE task, the nnU-Net encoder is used for the classification task.

|  | Backbone | Method | AUROC |
|---|---|---|---|
| CANDID | ResNet101 | Classification | 0.82 |
|  | ResNet50 | Seg-for-class | 0.91 |
| ISIC | ResNet101 | Classification | 0.59 |
|  | ResNet50 | Seg-for-class | 0.67 |

Table A7: Performance comparison when classification has a higher-capacity model (i.e., contains more parameters) compared to segmentation.

a higher capacity classification model—Resnet101, which has more parameters than the Resnet50 segmentation model—in Table A7. Again, we see improved performance with segmentation.

Finally, we emphasize that we matched training procedure for classification and segmentation networks, including matching the input data, augmentations, and codebase (which standardizes model checkpointing, loss function, hyperparameter tuning, etc.). Together, these results show it is not just model capacity or training procedure that lead to segmentation-for-classification's improved performance.

### A5.4 Additional performance metrics

In Table A8 we report additional performance metrics using standard classification and segmentation-for-classification (including semi-supervised methods) on the CANDID dataset.

### A5.5 Results on a natural image dataset

To show the proposed method extends to a natural image dataset, we include results classifying dog and cat breeds in the Oxford Pets dataset in the limited data regime (50 training images per class). We trained these models using ResNet50 architectures and the rule-based summarizing function. We observe segmentation-for-classification achieves an AUROC of 0.96 while standard classification achieves an AUROC of 0.81. These results show segmentation-for-classification improves average AUROC by 18.5% on a natural image dataset, mirroring our results on medical imaging datasets.

### A5.6 Results on multiclass datasets

While we focus on single-class tasks in the main text, we show here that segmentation-for-classification can directly be used for multiclass tasks. Instead of having a binary mask output from the segmentation network, a user simply needs to specify multiple output channels. First, we note the Oxford Pets dataset contains 37 classes of dog and cat breeds; as shown in the previous subsection, we see segmentation outperforms classification by 18.5% on this multiclass dataset. Further, we extend one of our medical imaging datasets to the multiclass setting and perform 3-class classification on ISIC, classifying lesions as benign nevi, melanoma, or seborrheic keratosis. Again, we see improved performance with segmentation-for-classification, which achieves a balanced AUROC of 0.61 compared to classification's 0.50 in the limited data regime.

| | # images with seg and class labels | # images with only class labels | # images with no labels | AUROC | Bal. Accuracy | Recall | Precision | Specificity |
|---|---|---|---|---|---|---|---|---|
| Classification (limited) | 0 | 198 | 0 | 0.74 (0.72 - 0.76) | 0.68 (0.66 - 0.70) | 0.60 (0.56 - 0.64) | 0.31 (0.29 - 0.34) | 0.76 (0.74 - 0.77) |
| Seg-for-class (limited) | 198 | 0 | 0 | 0.86 (0.84 - 0.88) | 0.78 (0.76 - 0.80) | 0.75 (0.71 - 0.78) | 0.42 (0.39 - 0.45) | 0.81 (0.80 - 0.82) |
| Seg-for-class, semi-sup | 198 | 0 | 3780 | 0.89 (0.87 - 0.90) | 0.81 (0.79 - 0.83) | 0.75 (0.71 - 0.78) | 0.51 (0.48 - 0.55) | 0.87 (0.86 - 0.88) |
| Seg-for-class, boosted semi-sup | 198 | 3780 | 0 | 0.90 (0.89 - 0.92) | 0.83 (0.81 - 0.84) | 0.73 (0.70 - 0.77) | 0.62 (0.59 - 0.66) | 0.92 (0.91 - 0.93) |
| Classification (abundant) | 0 | 3978 | 0 | 0.90 (0.89 - 0.92) | 0.82 (0.81 - 0.84) | 0.80 (0.77 - 0.83) | 0.49 (0.46 - 0.52) | 0.85 (0.84 - 0.86) |
| Seg-for-class (abundant) | 3978 | 0 | 0 | 0.96 (0.96 - 0.97) | 0.90 (0.89 - 0.92) | 0.89 (0.86 - 0.92) | 0.67 (0.63 - 0.70) | 0.92 (0.91 - 0.93) |

Table A8: Comparison of classification vs. semi-supervised segmentation-for-classification performance on the CANDID dataset in the most limited and abundant training data conditions. These models were trained with a balanced dataset. We report the average of each metric over the test set with 95% confidence intervals in parentheses.

## A6   Limitations

A limitation of this work includes the maximum sizes of the datasets. With much larger datasets, we expect overall classification performance to improve. However, the dataset sizes we use in this manuscript (2,000–20,000 for medical datasets; 100,000 for synthetic dataset) are not uncommon medical imaging dataset sizes, particularly for new applications or datasets specific to an institution. Further, we expect many benefits to hold in the very large data regime (e.g., robustness to spurious correlations, performance on very rare subsets, human assessment, and model evaluation). As an additional limitation, the segmentation-for-classification methods explored here are most applicable to abnormalities that can be segmented. Images for which even experts do not know which regions directly correspond to the class labels (e.g., some histopathology images [63]) have less obvious segmentation targets. In these situations, classification in combination with methods like GradCAM can help with feature discovery, for which segmentation-for-classification may be less well-suited. Similarly, classification labels which are more global in nature (e.g., age prediction from radiographs, atrophy of the brain in a head CT) do not have clear segmentation targets and may not benefit as much from the segmentation signal.

## A7   Broader impacts

This work focuses on automated medical image analysis. Automated image analysis algorithms have the potential to support radiologists by providing decision support, improving analysis efficiency, and improving analysis reproducibility. The work presented here helps to make progress towards these benefits by improving the performance of automated medical image analysis algorithms; improving the quality of model evaluation; and providing location information to the radiologist to facilitate human assessment. However, medical image analysis is a low failure tolerance application. The work presented here is meant as a contribution to shift the way we consider segmentation vs. classification algorithms in building medical image analysis tools; the specific neural networks trained here are not meant for clinical use.

