# OpenReview forum: "A case for reframing automated medical image classification as segmentation"
_NeurIPS.cc/2023/Conference — NeurIPS 2023 poster_

### Official Review · Reviewer_GK2b · 2023-07-02

**Soundness:** 2 fair
**Presentation:** 3 good
**Contribution:** 2 fair
**Rating:** 6
**Confidence:** 4

**Summary:**

This paper explores the benefits and drawbacks of using segmentation-based methods for classification tasks, particularly in medical imaging. This approach, known as segmentation-for-classification, has been shown to outperform traditional classification models, especially when the available dataset is small or when the classes are imbalanced. It can also handle rare subtypes better than conventional models.

**Strengths:**

- Improve aggregate performance: by enhancing the divergence between segmentation's positive and negative classes and increasing the quantity of annotation, segmentation-for-classification provides greater performance differences in situations with limited data, such as in small datasets, scenarios with low class prevalence, and rare subtypes.
- Reduced susceptibility to spurious correlations: Segmentation-for-classification is generally more robust to background features that are spuriously correlated with the target task, thus making it more reliable in classification tasks.
- Location information: Segmentation-for-classification inherently delivers location information, facilitating human assessment of the model's predictions. This location information is particularly useful in medical imaging, where abnormalities can be challenging to find.
- Comprehensive experiment: It is impressive to see the breadth of datasets used, covering synthetic data as well as three distinct medical datasets. The thorough exploration of different training regimes, including fully labeled, semi-supervised, and a boosted semi-supervised approach, greatly contributes to the depth and robustness of this investigation.

**Weaknesses:**

- Lack of Baseline Comparisons: While the study establishes that segmentation-for-classification performs better than traditional classification in certain scenarios, it does not benchmark these findings against other state-of-the-art methods for handling imbalanced datasets or for performing classification with limited data. Without these comparisons, it's hard to evaluate the true value of the proposed approach.
- Handling of Spurious Correlations: While the authors claim that segmentation-for-classification is less susceptible to spurious correlations, the empirical evidence provided seems limited. The specific mechanisms through which the proposed method avoids or mitigates these correlations could be elaborated on more clearly.

**Questions:**

Please address the questions mentioned in weakness.
Also, I am wondering how the segmentation-for-classification performs in 3D segmentation scenario.

**Limitations:**

Add more baseline comparison. Specific mechanism for mitigating spurious correlations. 3d data performance.

---

> ### Author Rebuttal · Authors · 2023-08-09
>
> Thank you for your suggestions and questions in your review. We were glad to hear the reviewer appreciated the many potential benefits of segmentation-for-classification Below, we answer your questions point-by-point. We would be glad for any additional discussion with or suggestions by the reviewer.
>
> *Q1: Lack of Baseline Comparisons: While the study establishes that segmentation-for-classification performs better than traditional classification in certain scenarios, it does not benchmark these findings against other state-of-the-art methods for handling imbalanced datasets or for performing classification with limited data. Without these comparisons, it's hard to evaluate the true value of the proposed approach.*
>
> We do use straightforward classification and segmentation networks in the original submission, as we were interested in exploring fundamental differences between the two task framings. We observed that segmentation-for-classification is able to achieve higher performance in the limited data regime (including with imbalanced datasets), improved robustness to spurious correlations, and other benefits without any sophisticated methods—simply by using a segmentation model, one is able to attain these benefits. We agree with the reviewer that additional classification baselines for these settings exist, but we note that they are different for different objectives (e.g., methods for reducing reliance on spurious correlations are different from methods that handle class imbalance are different from methods that improve interpretability). We believe the fact that segmentation handles all of these applications without specialized tooling is a major strength of the proposed approach.
>
> To address the reviewer’s concern, we are happy to include additional references and discussion of prior work in classification aimed at addressing these applications, for example [1, 2, 3]. Finally, we note that the SPINE dataset is balanced and we still observe improvements using segmentation-for-classification.
>
>
> *Q2: Handling of Spurious Correlations: While the authors claim that segmentation-for-classification is less susceptible to spurious correlations, the empirical evidence provided seems limited. The specific mechanisms through which the proposed method avoids or mitigates these correlations could be elaborated on more clearly.*
>
> We will add the mathematical statement and support (with proper reference to prior work) of why we expect segmentation-for-classification to be more robust to spurious correlations to the Appendix in Section A3, along with additional intuition and discussion elaborating on this point. Intuitively, spurious features carry less information about segmentation labels than image-level labels. Empirically, in the original submission we observed improved robustness with the CANDID dataset, where we saw performance on a difficult subset of patients improve from 0.58 to 0.84 AUROC. We also swept the strength of the spurious correlation in Figure 4(f) with the synthetic dataset, which showed classification’s performance dropoff with the increasing spurious correlation strength.
>
>
> *Q3: Also, I am wondering how the segmentation-for-classification performs in 3D segmentation scenario.*
>
> The SPINE dataset is a 3D dataset, for which we use 3D convolutional networks. We see similar trends as we saw in 2D. We note there isn’t anything in our analysis or method that is specific for 2D segmentation, supporting the observed results that hold in both 2D and 3D.
>
> Additionally, during this rebuttal period we have generated new results for a natural image dataset and for multiclass datasets. Both of these new applications worked out-of-the-box with the method described in the paper, suggesting the method and findings extend to additional datasets and settings the reviewer may be interested in; we provide details in the global response to reviewers.
>
>
> [1] Sohoni, N., Dunnmon, J., Angus, G., Gu, A., & Ré, C. (2020). No subclass left behind: Fine-grained robustness in coarse-grained classification problems. Advances in Neural Information Processing Systems, 33, 19339-19352.
>
> [2] Liu, E., Haghgoo, B., Chen, A., Raghunathan, A., Wei, P., Sagawa, Shiori., Liang, Percy., Finn, Chelsea. (2021.) "Just train twice: Improving group robustness without training group information", Proc. Int. Conf. Mach. Learn., pp. 6781-6792.
>
> [3] Azizi, S., Mustafa, B., Ryan, F., Beaver, Z., Freyberg, J., Deaton, J., ... & Norouzi, M. (2021). Big self-supervised models advance medical image classification. In Proceedings of the IEEE/CVF international conference on computer vision (pp. 3478-3488).

---

> > ### Comment · Reviewer_GK2b · 2023-08-15
> >
> > Thanks for your rebuttal and my concerns have been addressed. I would like to keep my rating

---

> > > ### Author Response · Authors · 2023-08-15
> > > **Final comment to reviewer GK2b**
> > >
> > > Thank you for your time reviewing and for your suggestions, we are glad we were able to answer your questions.

---

### Official Review · Reviewer_RFD8 · 2023-07-03

**Soundness:** 3 good
**Presentation:** 2 fair
**Contribution:** 3 good
**Rating:** 7
**Confidence:** 5

**Summary:**

This paper provides an intriguing and somewhat disruptive approach to medical image classification tasks. It implies that due to advancements in weakly-supervised, self-supervised, and semi-supervised segmentation techniques, the historical inclination towards image classification due to ease of training and label acquisition might be reconsidered. The authors argue for a shift towards segmentation, traditionally a more complex task, and demonstrate this by implementing "segmentation-for-classification" models. Using three retrospective datasets, they show these models outperforming traditional classification techniques in sample efficiency, performance on low-prevalence classes and rare subgroups, robustness to spurious correlations, and interpretability. However, the authors limit their consideration to tasks where the class of interest can be localized.

**Strengths:**

* The paper presents an innovative idea that can potentially change the current paradigm. If the results withstand, this reviewer sees a strong argument in saying that all existing segmentation models can easily be turned into powerful classification models by adding a small g() function at the end.
* The simplicity of the language used in the paper aids understanding.
* Figure 2 does an excellent job of visualizing the difference between classification and segmentation problems in the medical domain.

**Weaknesses:**

1. The most important issue this reviewer has discovered is how readily the authors dismissed the cost of providing pixel-level annotation. The difference between pixel-level and image-level (i.e. classes) annotation can be orders of magnitude. For example, in the study, this reviewer has been part of, an expert spent 279 s on average to produce pixel-level annotation for a single image, while it took only 2 s to generate a class label. Therefore, even a need to produce pixel-level annotations for even an absolute minority of images may be devastating. So far, from the literature and experience, it seems that such data is still needed, albeit in smaller quantities.
2. It is also important to point out a worry related to the study's design. In this work, the performance of augmented segmentation models is compared to the classification models, but no apparent effort is taken to ensure that the results obtained are not due to segmentation models being more powerful (which is often the case). Even the simplest segmentation architecture with the most naïve thresholding may have more parameters than the corresponding classification model. This is important to comment on in the main paper.
3. This brings another important observation - a lot of critical content, such as related works and algorithm details for summarising the probability maps, how multi-layer networks were trained and what architectures were used, the results for 4.2.2 etc., are relegated to the appendix, while arguably less crucial content occupies the main body of the text. This hampers the logical flow and comprehensiveness of the paper. For example, the discussion on the inherent differences between segmentation and classification, though interesting, feels redundant as it seems obvious that different methods would yield different results. This space could be better utilized to expand on related work and other methodological bits mentioned above. Section 5 can be greatly compressed, as it seems repetitive and less important.
4. Most figure captions are insufficiently informative, requiring the reader to refer back to the main text. Figure captions should be independent of the rest of the text.
5. The presentation of the results can be expended, with key information (e.g., number of true positives and false negatives) at this point missing.
6. The last two sections (4 and 5) feel more rushed and less systematic, reducing the overall readability of the paper.

**Questions:**

1. How do the authors justify the much higher cost and time commitment associated with pixel-level annotation as opposed to class-level annotation? Given the substantial difference, could the authors provide insights on the practicality of their approach in real-world scenarios?
2. Is it possible that the improved performance of the segmentation models over the classification models is simply due to them being inherently more powerful, possibly because they have more parameters? Could the authors elaborate on this aspect and discuss how they ensured a fair comparison between the two models?
3. Could the authors elaborate on the decision to relegate critical content to the appendix? Would it not improve the logical flow and comprehensiveness of the paper if key aspects such as related works, algorithm details, and results were included in the main text? Conversely, could parts of the discussion on the differences between segmentation and classification be moved to the appendix to make space for the aforementioned content in the main text?
4. Have the authors considered re-running their results multiple times to ensure they are reproducible?



**Limitations:**

The authors acknowledge their consideration only for classification/segmentation tasks where the class of interest can be localized. They do not address situations where the entire image must be evaluated, which is a significant limitation. Additionally, while the authors advocate for segmentation, they fail to adequately address the issue of pixel-level annotation's increased complexity relative to image-level annotation.

The paper does not discuss any potential negative societal impacts.

---

> ### Author Rebuttal · Authors · 2023-08-09
>
>
> Thank you for your careful read of our paper and many suggestions, which strengthened our submission. Below we respond to your points and describe how we updated our manuscript in response. We would be happy for further discussion or to answer any additional questions.
>
> *Q1: How do the authors justify the much higher cost…associated with pixel-level annotation?*
>
> We agree with you, per-image annotation cost is a major difference between classification and segmentation. Our goal with this paper is to better flesh out the decision space between segmentation vs. classification—it is not only the annotation cost that is different, but fundamental network behavior is different. As we show, segmentation-for-classification can lead to more performant, robust, and interpretable models. In clinical settings, one may prefer more performant or reliable models despite the added annotation cost.
>
> We do explore ways to ameliorate the annotation cost. We show in Section 4.2.2 that by using semi-supervision methods, we can use unlabeled data and <10% of the training data labeled with segmentation masks to achieve the performance of a model trained with 100% of the data labeled with classification labels. Further, we believe the landscape of image labeling is changing. In the last six months, the Segment Anything Model [1], SEER [2], and medical-imaging-specific extensions [3, 4] have been published. This changing landscape is part of the motivation for this submission: we want to consider the tradeoffs of segmentation vs. classification as tools that make labeling less burdensome become increasingly available.
>
> That being said, we do not mean to dismiss the high annotation cost required to produce segmentation labels. To address the reviewer’s concerns, we will adjust paragraphs 2 and 4 in the Introduction to more carefully position the paper with respect to the labeling burden of segmentation. Further, we will expand the discussion of labeling cost in Section 5 with more space given to comparative labeling burdens per image.
>
> [1] Kirillov, A., et al. (2023). Segment anything. arXiv:2304.02643.
>
> [2] Zou, X., et al. (2023). Segment everything everywhere all at once. arXiv:2304.06718.
>
> [3] Butoi, V. I., et al. (2023). Universeg: Universal medical image segmentation. arXiv:2304.06131.
>
> [4] Wang, C., et al. (2023). SAM-MED: A medical image annotation framework based on large vision model. arXiv:2307.05617.
>
> *Q2: Is it possible that the improved performance of the segmentation models over the classification models is simply due to them being inherently more powerful, possibly because they have more parameters?*
>
> Thank you for bringing this up, it is important to clarify. Please see the global response to all reviewers where we include two new tables showing that the improved performance is not simply due to increased model parameters.
>
> *Q3: Could the authors elaborate on the decision to relegate critical content to the appendix?*
>
> Thank you for the feedback. We had trouble fitting the paper into the 9 page limit, and as a result moved a lot of content to the Appendix. As we understand, there is an additional page allowed for the camera ready. With that in mind, we plan on making the following changes based on your (and other reviewers’) feedback and we are open to additional suggestions:
> - Bring details from Section A2 into the main text in Section 3, including:
>      - Algorithms 1 and 2 (combined into one algorithm box).
>      - Description of summarizing function architectures, including additional descriptions in Section 3 (moved from their current location in Section A2) and depicting the summarizing functions in a new Figure 3.
> - Expand the “Training” subsection in Section 4 to include model architectures, loss function, and optimizer (currently in Section A5.2).
> - Move results Table A6 to Main Text.
> - Expand the Summary of Related Work in the Introduction to be more thorough, while keeping the full Related Works in the Appendix. We agree the Related Works (currently located in the Appendix) is an important section, but it is lengthy (~1.3 pages) in order to be comprehensive, so we do not believe we can put the entire Related Works in the main text.
> - To make space for these changes, we can compress Sections 5 and 6 as well as remove Figure 2 and related references to the Figure.
>
> *Q4: Most figure captions are insufficiently informative, requiring the reader to refer back to the main text. Figure captions should be independent of the rest of the text.*
>
> Thank you for this suggestion, we will update all figure captions to be more comprehensive.
>
> *Q5: The presentation of the results can be expended, with key information (e.g., number of true positives and false negatives) at this point missing.*
>
> Thank you for the suggestion. In the original submission, we have additional performance metrics (balanced accuracy, recall, precision, specificity) in Table A6 for the CANDID dataset, which are indicative of performance beyond the AUROC. In the revised submission, we will include similar tables for the ISIC and SPINE datasets along with raw TP, TN, FP, FN numbers.
>
> *Q6: Have the authors considered re-running their results multiple times to ensure they are reproducible?*
>
> We are including new results for the CANDID segmentation and classification tasks with different random initializations in the limited data regime (Table R5). We observe a standard deviation of 0.0024 for segmentation-for-classification and 0.0054 for classification AUROC. In this rebuttal period we do not have time to rerun every experiment, but we can continue running experiments to have multiple seeds for the camera ready, if accepted.
>
> Additionally, due to reviewer D8HP’s suggestion, we trained multiclass and natural image models with the proposed method. Both of these applications worked out-of-the-box, suggesting the method and findings extend to additional datasets and settings (details in the global response).

---

> > ### Comment · Reviewer_RFD8 · 2023-08-14
> >
> > Dear Authors,
> >
> > Thank you for your comments; you have addressed most of my concerns.

---

> > > ### Author Response · Authors · 2023-08-14
> > > **Final comment to Reviewer RFD8**
> > >
> > > Great, we are glad we were able to address your concerns. Thank you again for your careful read and many suggestions!

---

### Official Review · Reviewer_rmFZ · 2023-07-06

**Soundness:** 4 excellent
**Presentation:** 3 good
**Contribution:** 3 good
**Rating:** 6
**Confidence:** 4

**Summary:**

The paper describes a set of insights obtained when using segmentation networks for a classification task. Classification was the task of choice due to issues with obtaining appropriate segmentation labels. However, with wider availability of datasets with appropriate labels, this is no longer the case. To facilitate consideration of using segmentation or classification networks, the paper presents a set of best practices and tradeoffs.

**Strengths:**

The manuscript is very thorough.

An important problem and best practices to solve it are discussed. In particular, the insights benefit the study of smaller datasets, rare subgroups, and robustness, all of which are important concerns in medical image analysis.

The supplementary material is very thorough and includes analysis of a synthetic dataset and three real datasets (X-ray, CT, skin).

Finally, some theoretical analysis evaluating separability of positive and negative segmentation and classification classes is provided.

I believe the analysis provided in the paper would be helpful to those trying to design an analysis task on medical image data.


**Weaknesses:**

After reading through the manuscript and supplementary material, I believe that some of the contents of the supplementary material (e.g., visualizations, choice of datasets) and the description of the pipeline can be summarized in a more detailed overview Figure (instead of Figure 3). While a lot of work (e.g., comparisons across networks) has been put into the project, not all of it is clearly explained in the main manuscript.

**Questions:**

Have the authors evaluated the effect of network size (# of parameters) vs performance? It seems like a smaller classification network and a larger segmentation network should in theory learn the same information.

Several weaknesses of classification networks and associated strengths of segmentation networks are discussed (e.g., in Figure 4, segmentation nearly always outperforms classification). When is the reverse true? Would there be cases where segmentation networks more easily overfit?


**Limitations:**

The paper presents several interesting insights into performance of classification via segmentation vs classification networks. However, I am not sure if the presented conclusions will generalize to other datasets/modalities. The paper could do a better job in clarifying the extent of its contributions.

---

> ### Author Rebuttal · Authors · 2023-08-09
>
> Thank you for your time reviewing our manuscript and for your suggestions. We were glad to hear you thought the work was thorough, addresses an important problem, and would be helpful to future readers. We were also glad to receive your comments and suggestions for improving the manuscript. Below, we answer your questions and describe how we have improved our submission as a result of your comments. We would be happy to answer any additional questions or hear additional suggestions.
>
>
> *Q1: After reading through the manuscript and supplementary material, I believe that some of the contents of the supplementary material (e.g., visualizations, choice of datasets) and the description of the pipeline can be summarized in a more detailed overview Figure (instead of Figure 3). While a lot of work (e.g., comparisons across networks) has been put into the project, not all of it is clearly explained in the main manuscript.*
>
> Thank you for this suggestion. We had trouble with the page limit and as a result placed a lot of content in the Appendix. We understand there is an additional page allowed for the camera ready. With that in mind, we plan on making the following changes and are open to additional suggestions:
> - Bring details from Section A2 into the main text in Section 3, including:
>      - Algorithms 1 and 2 (combined into one algorithm box).
>      - Description of summarizing function architectures, including additional descriptions in Section 3 (moved from their current location in Section A2) and depicting the summarizing functions in a new Figure 3. If the reviewer had specific content in mind they think would be useful to include in a modified Figure 3, we would appreciate their input!
> - Expand the “Training” subsection in Section 4 to include model architectures, loss function, and optimizer (currently in Section A5.2).
> - Move Table A6 to Main Text.
> - Expand the Summary of Related Work in the Introduction to be more thorough, while keeping the full Related Works in the Appendix.
> - To make space for these changes, we can compress Sections 5 and 6 as well as remove Figure 2.
>
>
> *Q2: Have the authors evaluated the effect of network size (# of parameters) vs performance?*
>
> This is a good question and we appreciate the suggestion to clarify this point. Please see our new results to answer this question in the global response, which we will include in the Appendix of the revised manuscript.
>
>
> *Q3: Several weaknesses of classification networks and associated strengths of segmentation networks are discussed (e.g., in Figure 4, segmentation nearly always outperforms classification). When is the reverse true? Would there be cases where segmentation networks more easily overfit?*
>
> This is an interesting question and one we investigated, though we did not find (either empirically or theoretically) settings within our defined scope (set in Section 2.1) in which segmentation-for-classification achieved worse performance than traditional classification. It is true that with very small datasets segmentation-for-classification models overfit, but the traditional classification networks were also not able to learn a more performant decision boundary in these settings.
>
> Outside of the scope we set, for example for tasks which require classifying the global image (e.g., classifying imaging modality), classification may achieve higher performance out-of-the-box than the segmentation-for-classification networks we use in the paper. Future work may investigate how to leverage segmentation networks for these global tasks; we will add discussion on this direction in our future work section. The future work is currently in Appendix A7, but we can move the future work paragraph into the main text’s Conclusion.
>
>
> *Q4: The paper presents several interesting insights into performance of classification via segmentation vs classification networks. However, I am not sure if the presented conclusions will generalize to other datasets/modalities. The paper could do a better job in clarifying the extent of its contributions.*
>
> We have included new results in the global response showing that the proposed method can be directly used in additional settings, including natural images and multiclass settings. We will add the multiclass and natural image results to the Appendix of the revised manuscript.
>
> To summarize, through our original experiments and this rebuttal, we have shown that the presented conclusions extend to different imaging modalities (RGB images, CT, x-ray), natural and medical images, 2D and 3D images/networks, and single class and multi class tasks. We have also explored trends thoroughly with synthetic datasets (Figure 4), showing settings in which performance differences emerge. Further, our theoretical analysis in Section 2 helps explain and support why one should expect the observed benefits to transfer to other datasets. Please let us know if there are additional generalizations or clarifications you are interested in discussing further.

---

> > ### Comment · Reviewer_rmFZ · 2023-08-15
> > **thank you**
> >
> > Thank you for the comments. The additional results helped clarify my concerns and I raised my rating.
> >
> > I believe papers that provide exploratory analysis of tasks such as the one presented are just as helpful as those with a specific methodological contribution.

---

> > > ### Author Response · Authors · 2023-08-15
> > > **Final comment to reviewer rmFZ**
> > >
> > > Thanks so much for your review and commentary, we are glad we were able to address your questions.

---

### Official Review · Reviewer_D8HP · 2023-07-06

**Soundness:** 2 fair
**Presentation:** 2 fair
**Contribution:** 2 fair
**Rating:** 4
**Confidence:** 3

**Summary:**

In this work, the authors explored the applications of deep learning in radiology, specifically focusing on image classification and segmentation tasks. The authors investigated the performance differences between classification and segmentation models on the same dataset and task using an information theoretic approach. They proposed a method called "segmentation-for-classification" that utilizes segmentation models for medical image classification. The authors compared this approach with traditional classification on three retrospective datasets, consisting of 2,018 to 19,237 samples. Through their analysis and experiments, they highlighted the benefits of segmentation-for-classification, including improved sample efficiency, better performance with fewer labeled images (up to 10 times fewer), particularly for low-prevalence classes and rare subgroups (up to 161.1% improved recall). They also demonstrated improved robustness to spurious correlations (up to 44.8% improved robust AUROC), as well as enhanced model interpretability, evaluation, and error analysis.


**Strengths:**

Through theoretical analysis, the benefits of using segmentation for learning classification datasets are validated, and extensive analysis and validation are conducted on small datasets.


**Weaknesses:**

1.The proposed method is not specifically designed for medical images; it is a general method but lacks validation in some common settings.
2.The method proposed in the paper is limited to binary classification requirements, which limits its scalability.
3.The method does not introduce a new module specifically designed to extend segmentation for classification tasks, lacking novelty.


**Questions:**

As shown in Figure 4, why is there such a significant difference between segmentation and classification on small datasets? Are the model parameters and training processes for segmentation and classification strictly controlled to be consistent?



**Limitations:**

Just as mentioned in the weakness section, the method proposed in the paper is limited to binary classification.

---

> ### Author Rebuttal · Authors · 2023-08-09
>
> Thank you for your suggestions and questions about our paper. Below, we respond to each of your points to answer your questions, provide new results, and describe how we are updating our submission in response to your comments. We would be happy to answer any additional questions or hear more suggestions during the discussion.
>
>
> *Q1: The proposed method is not specifically designed for medical images; it is a general method but lacks validation in some common settings.*
>
> In the global response, we clarify why we focus on medical images as a strong motivating application space. That being said, we agree with the reviewer that the methods can extend to natural images and show segmentation-for-classification improves average AUROC by 18.5% on a natural image dataset in the limited data regime. We will add these natural image results to the Appendix.
>
>
> *Q2: The method proposed in the paper is limited to binary classification requirements, which limits its scalability.*
>
> The method is not limited to binary classification; it can directly be used for multiclass settings. Instead of having a binary mask output from the segmentation network, a user simply needs to specify multiple output channels. We provide new experiments on multiclass data in the global response, showing segmentation-for-classification improves average AUROC compared to traditional classification by 18.5% in a natural image multiclass setting and by 22.0% in a medical image multiclass setting.
>
> To make sure this misunderstanding does not arise for future readers, we will describe the multiclass case in Section 3 and include the multiclass results in Section A5.
>
>
> *Q3: The method does not introduce a new module specifically designed to extend segmentation for classification tasks, lacking novelty.*
>
> Our contributions are in explaining why and when we see performance boosts using segmentation-for-classification (supported by our theoretical analysis), providing best practices for using segmentation-for-classification, and showing the number of tradeoffs between a traditional classification vs. segmentation-for-classification task framing.
>
> To develop best practices, we do test eight modules designed to extend segmentation for classification, but find that the simplest deterministic method works best (Figure 5). We believe this simple solution will help boost the usability of segmentation-for-classification, as practitioners can implement it with no extra computational cost or tuning costs as is required by more complex modules. Additionally, we do propose and evaluate a boosted semi-supervision method (Section 4.2.2), which has not been shown before and demonstrates how to adapt semi-supervised segmentation to the segmentation-for-classification setting.
>
>
> *Q4: As shown in Figure 4, why is there such a significant difference between segmentation and classification on small datasets?*
>
> Our analysis in Section 2.2 suggests this is because segmentation has a higher KL divergence between classes. Since the data is more separable using a segmentation task framing, the decision boundary between the two classes can be learned with fewer data points—leading to improved performance in the limited labeled data regime.
>
>
> *Q5: Are the model parameters and training processes for segmentation and classification strictly controlled to be consistent?*
>
> Thank you for bringing up this point, we agree it is important to clarify. Please see the global response for details on this question and new results on model capacity and performance. We will add these details to the revised manuscript clarifying this point for future readers in Section A5.

---

> ### Comment · Area_Chair_Tu7d · 2023-08-18
> **Please acknowledge reading the rebuttal.**
>
> Dear reviewer,
>
> Please acknowledge reading the rebuttal. One can acknowledge reading the rebuttal by posting an official comment on the open review platform.
>
> Best,
> AC

---

### Author Rebuttal · Authors · 2023-08-09

We thank the reviewers for their time and helpful suggestions, which helped us strengthen our submission. We were glad to hear the reviewers recognized the benefits and potential impact of segmentation-for-classification (reviewers rmFZ, RFD8, GK2b), found the paper thorough (reviewers D8HP, rmFZ, GK2b), and considered the approach innovative (reviewer RFD8). In our global response, we respond to common questions and report additional experiments. In the individual responses, we address each reviewer’s questions point-by-point.

**Method generality.** Reviewers D8HP, GK2B, and rmFZ asked about the method's generality, both within medical imaging and in other areas. First, we highlight our intentional focus on medical imaging. Then, we present additional experiments requested by the reviewers showing the method’s applicability to natural, multiclass, and 3D images.

We focus on medical images as segmentation-for-classification helps address problems that are particularly acute in this setting:
- Many medical image datasets collected from within an institution are small or contain rare subgroups [1, 2, 3]; we show segmentation-for-classification improves performance in this limited data regime (e.g., up to 16.2% improved aggregate performance; up to 161.1% improved recall on a rare subset).
- Sensitivity to spurious correlations is particularly worrisome in low-failure-tolerance applications like medical imaging [1]. We’ve observed segmentation improves robust AUROC up to 44.8%.
- Segmentation inherently delivers location information about abnormalities, helping clinicians interpret the results of the classifier in clinical workflows.

Thus, we believe medical image analysis is a strong motivating setting and well-suited application space for segmentation-for-classification. The improved performance and reliability in this risk-intolerant setting as well as the close alignment with clinical workflows helps justify the added cost of segmentation, although we note recent work in self-supervision and foundation models helps ameliorate the cost of labeling segmentation data [4, 5, 6].

That being said, we agree with reviewer D8HP that the methods extend to natural images and may be of general interest. To show the proposed methods extend to a natural image dataset, we include new results in Table R1 classifying dog and cat breeds in the Oxford Pets dataset in the limited data regime (50 training images per class). We see segmentation-for-classification improves average AUROC by 18.5%. We will include these results in the revised Appendix.

To show the method applies to multiclass settings (reviewer D8HP), the Oxford Pets dataset contains 37 classes of dog and cat breeds; as above, we see segmentation outperforms classification by 18.5%. Further, we extend one of our medical imaging datasets to the multiclass setting and perform 3-class classification on ISIC. Again, we see improved performance with segmentation-for-classification (Table R2). We will describe the multiclass case in Section 3 and include the multiclass results in Section A5.

To show the method applies to 3D data, we emphasize the SPINE dataset in our original submission is a 3D dataset, and we observe the same trends in 3D as we do in 2D.

**Model capacity.** Reviewers D8HP, rmFZ, and RFD8 ask about differences in model capacity between the classification and segmentation. We first clarify how we selected architectures in the original submission. Then, we present new results showing that model capacity does not explain the observed performance differences.

First, we clarify that in the original paper we tested multiple architectures for each task and dataset (Table A4). We did this to give each task and dataset the best performance, as it’s not clear the best architecture for segmentation on a given dataset is the same that is best for classification.

To address the reviewer’s questions, we present two new results. In Table R3 we use the same backbone for both segmentation and classification. We use Resnet50 as the 2D backbone as it is a common backbone known to be useful for classification and segmentation. We see that segmentation shows the same or greater performance enhancement than what appeared in the original paper. We further note that all of our synthetic experiments were run with a standardized backbone.

However, even when using the same backbone, segmentation models still have more parameters due to the convolutional decoder (as noted by reviewer RFD8). We next provide results comparing performance using a higher capacity classification model—Resnet101, which has more parameters than the Resnet50 segmentation model—in Table R4. Again, we see drastically improved performance with segmentation.

Finally, we clarify that the training procedure is matched for classification and segmentation, including matching the input data, augmentations, and codebase (which standardizes model checkpointing, loss function, hyperparameter tuning, etc.).

Together, these results show it is not model capacity or training procedure that leads to segmentation-for-classification’s improved performance. We will add these new details to the revised manuscript for future readers.


[1] Oakden-Rayner, L., et al. (2020). Hidden stratification causes clinically meaningful failures in machine learning for medical imaging. ACM CHIL.

[2] Ng, D., et al. (2021). Federated learning: a collaborative effort to achieve better medical imaging models for individual sites that have small labelled datasets. Quantitative Imaging in Medicine and Surgery.

[3] Varoquaux, G., et al. (2022). Machine learning for medical imaging: methodological failures and recommendations for the future. NPJ DM.

[4] Kirillov, A., et al. (2023). Segment anything. arXiv:2304.02643.

[5] Zou, X., et al. (2023). Segment everything everywhere all at once. arXiv:2304.06718.

[6] Butoi, V. I.,  et al. (2023). Universeg: Universal medical image segmentation. arXiv:2304.06131.

---

### Decision · Program_Chairs · 2023-09-21

**Decision:**

Accept (poster)

**Comment:**

The submission was reviewed by four reviewers. The reviewers agree that the ideas presented in the paper are interesting, well validated and worth publishing. AC notices that one of the reviewer that recommends to reject did not engage in the discussion with the authors. AC concludes that the criticisms of this reviewer were resolved by the authors' response.  Thus, AC agrees with the majority of the reviewers and recommends to accept the submission.